# Collaborative Learning by Detecting Collaboration Partners

**Shu Ding, Wei Wang** [*]
National Key Laboratory for Novel Software Technology
Nanjing University, Nanjing 210023, China
`{dings, wangw}@lamda.nju.edu.cn`

## Abstract

Massive amounts of data are naturally dispersed over different clients in many real-world applications, collaborative learning has been a promising paradigm that allows to learn models through collaboration among the clients. However, leveraging these dispersed data to learn good models is still challenging since data over different clients are heterogeneous. Previous works mainly focus on learning the centralized model for all clients or learning a personalized model for each client. When there are numerous clients, the centralized model performs badly on some clients, while learning a personalized model for each client costs unaffordable computational resources. In this paper, we propose the collaborative learning method to detect collaboration partners and adaptively learn $K$ models for numerous heterogeneous clients. We theoretically prove that the model learned for each client is a good approximation of its personalized model. Experimental results on real-world datasets verify the effectiveness of our method.

## 1 Introduction

In many real-world applications, massive amounts of data are naturally dispersed over numerous clients and each client only has limited data. The model trained with local data on each client has poor performance since local data are insufficient. Leveraging these dispersed data on these clients to build well-generalized models is challenging. In recent years, collaborative learning has been a promising paradigm that enables the clients to collaboratively learn the models (Blum et al., 2017). Blum et al. (2017) formalized two settings of collaborative learning, *i.e.*, personalized setting and centralized setting. The personalized setting allows to return a model for each client, while the centralized setting only allows to return one single model for all clients. They evaluated the efficiency of collaborative learning with *overhead*, *i.e.*, the ratio between the sample complexity of collaborative learning and its non-collaborative counterpart, and proved that the overhead of personalized setting is $O(\ln N)$ and the overhead of centralized setting is $O(\ln^2 N)$. Later, Nguyen & Zakynthinou (2018) and Chen et al. (2018) proved that the overhead of centralized setting could be improved to $O(\ln N)$, which matches its lower bound $\Omega(\ln N)$.

Generally, the data on different clients come from heterogeneous distributions. For example, in the credit card fraud behavior detection task, the users' revenue and expenditure behavior and credit rating information are different among banks in regions with diverse socioeconomic characteristics. To deal with heterogeneous data, Mohri et al. (2019) proposed Agnostic Federated Learning which optimizes the performance of the model for the single worst client with the min-max optimization scheme. Mansour et al. (2020) and Deng et al. (2020) advocated that combining local and global models could improve the generalization ability. Nevertheless, the learned single model may perform badly on clients whose distributions are different from the average distribution over all clients.

---

[*]Corresponding author.

To achieve good performance, a personalized model should be learned for each client, but learning these personalized models is impractical when the number of clients $N$ is very large since this costs unaffordable computational resources. Intuitively, it's better to return $K$ ($1 \leqslant K \ll N$) appropriate models by analyzing the collaboration relationships among $N$ heterogeneous clients. We expect that the performance of the returned model is comparable to that of each client's optimal personalized model. Based on this intuition, in this paper we propose the collaborative learning method to detect collaboration partners for each client and adaptively learn $K$ models for $N$ heterogeneous clients. We first estimate each client's contribution based on the derived generalization bound, and then partition the clients into $K$ groups and return a model for each group. We also theoretically prove that the model returned for each client is a good approximation of its optimal personalized model. Experimental results on real-world datasets verify the effectiveness of our method.

## 2 Related Work

Blum et al. (2017) first theoretically studied collaborative learning and defined *overhead* as the ratio between the sample complexity of collaborative learning and its non-collaborative counterpart. They proved that the overhead of personalized setting is $O(\ln N)$ and the overhead of centralized setting is $O(\ln^2 N)$. Later, Nguyen & Zakynthinou (2018) and Chen et al. (2018) proved that the overhead of centralized setting could be improved to $O(\ln N)$ by using multiplicative weights methods, which matches its lower bound $\Omega(\ln N)$. When there are an $\eta$ fraction of malicious clients, Qiao (2018) proposed an algorithm that achieves $O(\eta N + \ln N)$ overhead, which is proved to be the worst-case optimal. Since the communication cost between clients and the central server is expensive, Blum et al. (2021) developed algorithms by using distributed boosting to reduce the communication cost without compromising the sample complexity.

To deal with data heterogeneity, Mohri et al. (2019) proposed Agnostic Federated Learning, which optimizes the performance of the model for the single worst client with the min-max optimization scheme. Mansour et al. (2020) proved that one single model generalizes badly on the clients whose distributions vary from the average distribution. Mansour et al. (2020) and Deng et al. (2020) advocated that the mixture of local and global models could improve the generalization ability. Li et al. (2020) highlighted the detrimental effects of heterogeneity by observing that heterogeneity leads to poor convergence rate. They proposed FedProx to tackle heterogeneity and provided convergence guarantee. Wang et al. (2020b) proposed the normalized averaging method FedNova to eliminate objective inconsistency while preserving fast convergence rate. Dinh et al. (2020) proposed a personalized algorithm pFedMe by using Moreau envelops as regularized loss functions and theoretically showed it could obtain quadratic speedup. Huang et al. (2020) proposed a novel framework FEDAMP which allows each client to collaboratively train its own model without using the global model and provided the convergence guarantee. These works mainly focus on learning the global model which performs well on the average distribution. However, it may perform badly on the clients whose distributions are different from the average distribution. In a concurrent work (Even et al., 2022), the authors proposed methods to achieve the tradeoff between personalization and coordination, which tackles the same problem in our paper from a different perspective.

Client selection has also been studied in some works. Considering that different groups of clients may have different learning objectives, Ghosh et al. (2020) proposed IFCA which alternately estimates the cluster identities of the clients and optimizes the models for each cluster. Fraboni et al. (2021) introduced clustered sampling with better client representativity to reduced variance of the clients stochastic aggregation weights and provided two cluster approaches based on sample size and model similarity. Tuor et al. (2020) proposed a method for selecting relevant data for model training, where they used the model trained on the benchmark dataset which is task-specific to evaluate the data relevance. Cui et al. (2022) studied a rational collaboration scheme called collaboration equilibrium where each client collaborates with certain members who maximally improve the model learning and isolates the others who make little contribution. However, these works are heuristic and lack theoretical support.

## 3 Preliminaries

Let $\mathcal{X}$ be the instance space and $\mathcal{Y} = \{0, 1\}$ is the label space. In the collaborative learning setting, there are $N$ clients $\{C_1, \ldots, C_N\}$ with distributions $\{\mathcal{D}_1, \ldots, \mathcal{D}_N\}$. In practice, client $C_i$ only has

access to $m_i$ examples $S_i = \left\{ \left( \boldsymbol{x}_1^i, y_1^i \right), \ldots, \left( \boldsymbol{x}_{m_i}^i, y_{m_i}^i \right) \right\}$ drawn from distribution $\mathcal{D}_i$, $i = 1, \ldots, N$. Let $M = \sum_{i=1}^N m_i$ be the total number of examples. We consider the hypothesis space $\mathcal{H}$ with VC dimension $d$. A hypothesis $h \in \mathcal{H}$ is a mapping $h : \mathcal{X} \to \mathcal{Y}$. Let $l$ be the loss function and $l(h(\boldsymbol{x}), y)$ is the loss of hypothesis $h \in \mathcal{H}$ on example $(\boldsymbol{x}, y)$. We assume that $l$ is upper bounded by a constant $\mu$. For client $C_i$, denote $\mathcal{L}_{\mathcal{D}_i}(h)$ as the expected risk of a hypothesis $h \in \mathcal{H}$ with respect to distribution $\mathcal{D}_i$ and $\hat{\mathcal{L}}_{S_i}(h)$ is the empirical risk of a hypothesis $h \in \mathcal{H}$ with respect to sample $S_i$ :

$$\mathcal{L}_{\mathcal{D}_i}(h) = \mathbb{E}_{(\boldsymbol{x}, y) \sim \mathcal{D}_i} [l(h(\boldsymbol{x}), y)], \quad \hat{\mathcal{L}}_{S_i}(h) = \frac{1}{m_i} \sum_{j=1}^{m_i} [l(h(\boldsymbol{x}_j^i), y_j^i)].$$

For client $C_i$, its goal is to pursue the model $h_i^\star$ that minimizes the expected risk on its local distribution $\mathcal{D}_i$, i.e., $h_i^\star = \arg\min_{h \in \mathcal{H}} \mathcal{L}_{\mathcal{D}_i}(h)$. Nevertheless, it is impractical to find $h_i^\star$ since $C_i$ only has access to limited $m_i$ examples. Thus the key incentive of clients $C_i$ participating in collaboration is the desire of seeking the model that could generalize well on its local distribution $\mathcal{D}_i$ with samples $\{S_1, \ldots, S_N\}$ from all the clients.

# 4   Our Work

In collaborative learning scenarios, the common strategy is to train the model over the weighted union of all samples, i.e., $S_{\boldsymbol{\alpha}} = \sum_{j=1}^N \alpha_j S_j$ (McMahan et al., 2017; Mohri et al., 2019). Here $\boldsymbol{\alpha} = (\alpha_1, \ldots, \alpha_N) \in \Delta^N$ where $\Delta^N$ is the $N$-simplex, i.e., $\sum_{j=1}^N \alpha_j = 1$ and $\alpha_j \geqslant 0$ for $j = 1, \ldots, N$. The weighted empirical risk $\hat{\mathcal{L}}_{\boldsymbol{\alpha}}(h)$ associated to hypothesis $h \in \mathcal{H}$ can be denoted as $\hat{\mathcal{L}}_{\boldsymbol{\alpha}}(h) = \sum_{j=1}^N \alpha_j \hat{\mathcal{L}}_{S_j}(h)$. In this way, the weighted empirical risk $\hat{\mathcal{L}}_{\boldsymbol{\alpha}_i}(h)$ for client $C_i$ can be denoted as $\hat{\mathcal{L}}_{\boldsymbol{\alpha}_i}(h) = \sum_{j=1}^N \alpha_{ij} \hat{\mathcal{L}}_{S_j}(h)$ with the corresponding collaboration vector $\boldsymbol{\alpha}_i = (\alpha_{i1}, \ldots, \alpha_{iN})$. The model for $C_i$ can be learned by minimizing $\hat{\mathcal{L}}_{\boldsymbol{\alpha}_i}(h)$. Let $\hat{h}_{\boldsymbol{\alpha}_i} = \arg\min_{h \in \mathcal{H}} \hat{\mathcal{L}}_{\boldsymbol{\alpha}_i}(h)$ be the hypothesis that minimizes the weighted empirical risk $\hat{\mathcal{L}}_{\boldsymbol{\alpha}_i}(h)$. We provide the generalization bound of $\hat{h}_{\boldsymbol{\alpha}_i}$ for client $C_i$ in the following Theorem 1.

**Theorem 1.** *Let $\mathcal{H}$ be the hypothesis space with VC-dimension $d$. Let $h_i^\star = \arg\min_{h \in \mathcal{H}} \mathcal{L}_{\mathcal{D}_i}(h)$ be the hypothesis that minimizes the expected risk on $\mathcal{D}_i$ and $\hat{h}_{\boldsymbol{\alpha}_i} = \arg\min_{h \in \mathcal{H}} \hat{\mathcal{L}}_{\boldsymbol{\alpha}_i}(h)$ is the hypothesis that minimizes the weighted empirical risk. For any given $\delta \in (0, 1)$ and $\forall i \in \{1, \ldots, N\}$, the following bound holds with probability at least $1 - \delta$:*

$$\mathcal{L}_{\mathcal{D}_i}(\hat{h}_{\boldsymbol{\alpha}_i}) - \mathcal{L}_{\mathcal{D}_i}(h_i^\star) \leqslant 2 \sum_{j=1}^N \alpha_{ij} d_{\mathcal{H}}(\mathcal{D}_i, \mathcal{D}_j) + 2\mu \sqrt{\sum_{j=1}^N \frac{\alpha_{ij}^2}{m_j}} \sqrt{8(d \log(2M) + \log \frac{8}{\delta})}. \quad (1)$$

*Here $d_{\mathcal{H}}(\mathcal{D}_i, \mathcal{D}_j) = \sup_{h \in \mathcal{H}} \left| \mathcal{L}_{\mathcal{D}_i}(h) - \mathcal{L}_{\mathcal{D}_j}(h) \right|$ is the Integral Probability Metrics (IPM) which measures the divergence between distributions $\mathcal{D}_i$ and $\mathcal{D}_j$.*

We leave the proof of Theorem 1 in Appendix A due to space limitation. Theorem 1 reveals that for client $C_i$ the gap between $\hat{h}_{\boldsymbol{\alpha}_i}$ and $h_i^\star$ depends on the weight $\boldsymbol{\alpha}_i$, the number of examples $m_i$ and the IPM divergence. It further suggests that the optimal model for client $C_i$ can be obtained by choosing appropriate $\boldsymbol{\alpha}_i$ to minimize the right-hand side of Eq.(1). The optimal $\boldsymbol{\alpha}_i^\star = (\alpha_{i1}^\star, \ldots, \alpha_{iN}^\star)$ for client $C_i$ can be derived with the following Theorem 2.

**Theorem 2.** *For client $C_i$, let $\Xi_i^j = d_{\mathcal{H}}(\mathcal{D}_i, \mathcal{D}_j)$ where $j \in \{1, \ldots, N\}$, $\forall i \in \{1, \ldots, N\}$. Given VC dimension $d$, total sample size $M$, constants $\mu$ and $\delta$, let $\lambda = \mu \sqrt{8(d \log(2M) + \log \frac{8}{\delta})}$. For client $C_i$, sort $\{\Xi_i^1, \ldots, \Xi_i^N\}$ in ascending order to get $\{\Xi_i^{\sigma(1)}, \ldots, \Xi_i^{\sigma(N)}\}$, i.e., $\Xi_i^{\sigma(1)} \leqslant \ldots \leqslant \Xi_i^{\sigma(N)}$, where $\sigma(j) \in \{1, \ldots, N\}$. The optimal $\boldsymbol{\alpha}_i^\star = \{\alpha_{i1}^\star, \ldots, \alpha_{iN}^\star\}$ for client $C_i$ is given by*

$$\alpha_{ij}^\star = \left[ \frac{m_j(\zeta - \Xi_i^j)}{\sum_{q \leqslant q_i} m_{\sigma(q)}(\zeta - \Xi_i^{\sigma(q)})} \right]_+. \quad (2)$$

*Here $[\cdot]_+ = max(\cdot, 0)$, $\zeta$ is the larger root of equation $\sum_{q \leqslant q_i} m_{\sigma(q)} \left( \zeta - \Xi_i^{\sigma(q)} \right)^2 = \lambda^2$, and $q_i = \arg\max_t \left\{ t \Big| \zeta \geqslant \Xi_i^{\sigma(t)} \wedge \left( \sum_{q \leqslant t} m_{\sigma(q)} \Xi_i^{\sigma(q)} \right)^2 \geqslant \left( \sum_{q \leqslant t} m_{\sigma(q)} \right) \left( \sum_{q \leqslant t} m_{\sigma(q)} (\Xi_i^{\sigma(q)})^2 - \lambda^2 \right) \right\}$.*

We leave the proof of Theorem 2 in Appendix B. Theorem 2 can be interpreted as follows. The optimal $\boldsymbol{\alpha}_i^\star$ makes the balance between the divergence and the number of examples. If $\alpha_{ij}^\star > 0$, then $C_j$ is beneficial to $C_i$ in the collaboration process, *i.e.*, $C_i$ can utilize the data on $C_j$ to relieve the scarcity of its local data. $\alpha_{ij}^\star$ represents the *contribution* that client $C_j$ makes to $C_i$. We use the directed graph $A = (V, E)$ to represent the collaboration relationships among clients, where $|V| = N$, node $i$ denotes client $C_i$ and the weight of edge from $j$ to $i$ is $\alpha_{ij}^\star$. Here we provide an example in Figure 1 to give a more concrete interpretation. If there exists an edge from node $j$ to node $i$, then $C_i$ needs the data from $C_j$. In Figure 1, the contribution that $C_2$ and $C_4$ need from all the clients are more similar than that of $C_4$ and $C_5$, *i.e.*, the incoming edges of $C_2$ and $C_4$ are more similar than that of $C_4$ and $C_5$. We could probably return the same model for $C_2$ and $C_4$ while it is inappropriate to return the same model for $C_4$ and $C_5$.

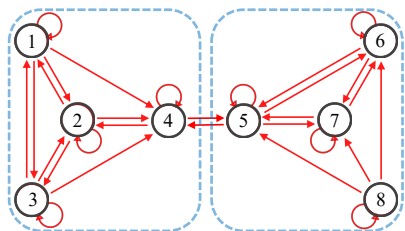

Figure 1: Illustrative example of the directed graph $A$. Node $i$ denotes client $C_i$ and the weight of edge from $j$ to $i$ is $\alpha_{ij}^\star$.

We refer $\hat{h}_{\boldsymbol{\alpha}_i^\star}$ with respect to the optimal $\boldsymbol{\alpha}_i^\star$ as the *personalized* model for client $C_i$. When the number of clients $N$ is large, learning a *personalized* model for each client is impractical since this costs unaffordable computational resources. An alternate way is to learn $K$ ($1 \leqslant K \ll N$) appropriate models which are expected to be comparable to the *personalized* models. Intuitively, if client $C_i$ and $C_j$ have similar incoming edges, then we can return the same model for them since they need similar contribution from other clients. In this way, $C_i$ and $C_j$ are called *collaboration partners*. This motivates us to partition $N$ clients into $K$ groups based on the idea that *collaboration partners* should be in the same group. If we set $K = 2$ in Figure 1, then the group structure is most likely to be $\{C_1, C_2, C_3, C_4\}$ and $\{C_5, C_6, C_7, C_8\}$ as shown in Figure 1.

### 4.1 Collaboration with Modularity Maximization

To evaluate the incoming-edge similarity among clients, we construct matrix $\mathbf{U}$ which is defined as

$$\mathbf{U} = \mathbf{D}_{in}^{-\beta} \mathbf{A}\mathbf{A}^T \mathbf{D}_{in}^{-\beta}, \tag{3}$$

where $\mathbf{A}$ is the adjacency matrix of the directed graph $A$, $\mathbf{D}_{in}$ is the diagonal matrix of in-degrees in graph $A$ and $\beta$ is the discounting parameter. We set $\beta = \frac{1}{2}$ as suggested in Satuluri & Parthasarathy (2011). Large $U_{ij}$ represents large similarity between clients, which provides the possibility of detecting the communities for the clients and partitioning collaboration partners into groups.

Modularity has been used as an objective function to evaluate the quality of group partitions by measuring the density of intra-group edges as compared to inter-group edges (Newman & Girvan, 2004), which is formulated as follows:

$$Q(\mathcal{G}) = \frac{1}{W} \sum_k^K W_{in}^{G_k} - \frac{1}{4W^2} \sum_k^K (W_{vol}^{G_k})^2. \tag{4}$$

Here $\mathcal{G} = \{G_1, \ldots, G_K\}$ is the group partition. $W = \frac{1}{2} \sum_i \sum_j w_{ij}$ is the sum of all edge weight. $W_{in}^{G_k} = \sum_{i,j \in G_k} w_{ij}$ is the total edge weight within group $G_k$ and $W_{vol}^{G_k} = \sum_{i \in G_k} \sum_j w_{ij}$ is the total edge weight attached to nodes in group $G_k$. Higher modularity implies better group partitions and modularity maximization has been widely used for detecting meaningful group partitions (Clauset et al., 2004; Newman, 2006a,b). Although maximizing modualarity is NP-hard, approximation guarantees can be obtained by introducing the technique of rounding semidefinite programs. We rewrite Modularity as follows:

$$Q(\mathcal{G}) = \frac{1}{2W} \sum_{i,j} \left[ w_{ij} - \frac{d_i d_j}{2W} \right] \delta(g_i, g_j), \tag{5}$$

where $d_i = \sum_j w_{ij}$ is the degree of vertex $i$, and $g_i$ is the group membership of client $C_i$ where $\delta(g_i, g_j) = 1$ if $g_i = g_j$ (i.e., client $C_i, C_j$ belong to the same group) and 0 otherwise. Let $\boldsymbol{e}_j \in \mathbb{R}^K$ be the unit vector with 1 in the $j$-th coordinate and 0 elsewhere. Let $\boldsymbol{\nu}_i$ be the variable that indicates the group membership of client $C_i$, i.e., $\boldsymbol{\nu}_i = \boldsymbol{e}_j$ means $C_i$ belongs to group $G_j$. Then the modularity maximization problem can be formulated as the following vector programming:

$$\max \quad \sum_{i,j} \left( \frac{w_{ij}}{2W} - \frac{d_i d_j}{4W^2} \right) \boldsymbol{\nu}_i \cdot \boldsymbol{\nu}_j \tag{6}$$

$$s.t. \quad \boldsymbol{\nu}_i \in \{\boldsymbol{e}_1, \ldots, \boldsymbol{e}_K\}, \quad \forall i \in \{1, \ldots, N\}.$$

Let $\mathcal{M}$ denote the modularity matrix with its elements $\mathcal{M}_{ij} = \frac{w_{ij}}{2W} - \frac{d_i d_j}{4W^2}$. Denote $\mathcal{M}^+ = \{(i,j) \mid \mathcal{M}_{ij} > 0\}$ and $\mathcal{M}^- = \{(i,j) \mid \mathcal{M}_{ij} < 0\}$. Noticing that $\sum_{ij} \mathcal{M}_{ij} = 0$, the above vector programming can be relaxed to the following SemiDefinite Program (SDP) by adding a constant $\sum_{\mathcal{M}^+} \mathcal{M}_{ij}$ to the objective and relaxing the constraints. The proof can be found in Appendix C.

$$\max \quad \sum_{\mathcal{M}^+} \mathcal{M}_{ij} \boldsymbol{\nu}_i \cdot \boldsymbol{\nu}_j + \sum_{\mathcal{M}^-} -\mathcal{M}_{ij} (1 - \boldsymbol{\nu}_i \cdot \boldsymbol{\nu}_j) \tag{7}$$

$$s.t. \quad \boldsymbol{\nu}_i \cdot \boldsymbol{\nu}_i = 1, \quad \forall i \in \{1, \ldots, N\},$$
$$\boldsymbol{\nu}_i \cdot \boldsymbol{\nu}_j \geqslant 0, \quad \forall i \neq j,$$
$$\boldsymbol{\nu}_i \in \mathbb{R}^K, \quad \forall i \in \{1, \ldots, N\}.$$

Let $\mathrm{OPT}_{Q(\mathcal{G})}$ be the maximum value of modularity defined on the similarity matrix $\mathbf{U}$. Dinh et al. (2015) provided the following result to guarantee the approximation property for the solution of the SDP in Eq.(7). Lemma 1 implies that we could find reasonable group partitions by solving the SDP in Eq.(7) for maximizing modularity on $\mathbf{U}$.

**Lemma 1.** *Given matrix $\mathbf{U}$, let $Q(\mathcal{G})$ be the modularity value of the group partition $\mathcal{G}$ obtained by solving the SDP in Eq.(7) using the rounding techniques in Charikar et al. (2003). Then $Q(\mathcal{G})$ satisfies $Q(\mathcal{G}) > \kappa \mathrm{OPT}_{Q(\mathcal{G})} - (1 - \kappa)$, where $\kappa = 0.766$ is the approximation factor.*

After obtaining the group partitions, we train models for the groups according to their properties. Since we assign the clients into $K$ groups, there may exist some clients which have small similarity with other clients in the same group. We should pay attention to these clients.

We call edge $e_{ij} \in \mathbf{U}$ the *weak edge* if its weight satisfies $w_{ij} < \frac{1}{N}$, else *strong edge* otherwise. Given group partition $\mathcal{G} = \{G_1, \ldots, G_K\}$, let $Z_{in}$ be the number of weak edges within all groups and $N_k$ is the number of clients in group $G_k$ where $\sum_{k=1}^K N_k = N$. For client $C_i$ in group $G_k$, we call $C_i$ the *bad client* if $C_i$ is isolated after removing all the weak edges within group $G_k$. Generally, we keep the largest part when a group is divided into several disjoint parts after removing all weak edges within this group, and clients that do not belong to the largest part are *bad clients*. Otherwise, we call it the *good client*. In Figure 1, $C_1, C_2, C_3, C_4$ are in the same group since they have relatively similar incoming edges, and $C_5, C_6, C_7$ are in the same group for the same reason. $C_8$ is most likely to be a bad client since it has less similar incoming edges with $C_5, C_6, C_7$ in the same group.

To guarantee the performance of the learned models, we pick out the bad clients since they have small similarity with other clients in the same group. We return $\hat{h}_{\boldsymbol{\alpha}_{G_k}}$ for clients in group $G_k$ which is obtained by minimizing $\hat{\mathcal{L}}_{\boldsymbol{\alpha}_{G_k}}(h)$ where $\boldsymbol{\alpha}_{G_k} = \frac{1}{|G_k \backslash \mathcal{B}|} \sum_{C_i \in G_k \backslash \mathcal{B}} \boldsymbol{\alpha}_i^\star$. We refer this method as **A**daptive **C**ollaborative **L**earning with **M**odularity **M**aximization (ACLMM). The whole process is summarized in Algorithm 1, where we use the rounding techniques introduced in Charikar et al. (2003) to solve the SDP in Eq.(7) for modularity maximization to obtain the group partition. Let $\mathcal{B}$ be the set of bad clients in the group partition returned by solving the SDP. We provide the following Lemma 2 to bound the number of bad clients.

**Lemma 2.** *Given the group partition $\mathcal{G} = \{G_1, \ldots, G_K\}$ returned by Algorithm 1, assume $N_k \geqslant 2\sqrt{Z_{in}}, \forall k \in \{1, \ldots, K\}$. Let $N_{min} = \min_k N_k$, then $|\mathcal{B}|$ satisfies $|\mathcal{B}| \leqslant \frac{N_{min} - \sqrt{N_{min}^2 - 4Z_{in}}}{2}$, where $Z_{in} \leqslant \frac{N}{2(N-1)} \left[ \frac{N^2 - KN}{K} - 2W \left( (\kappa+1)\mathrm{OPT}_{Q(\mathcal{G})} - \frac{K-1}{K} \right) \right]$ and $\kappa = 0.766$ is a constant.*

The proof of Lemma 2 can be found in Appendix C. Lemma 2 indicates that the number of weak edges within all groups $Z_{in}$ depends on the maximum modularity $\mathrm{OPT}_{Q(\mathcal{G})}$ and the number of

---

**Algorithm 1** Adaptive Collaborative Learning with Modularity Maximization (ACLMM)

1: **procedure** TRAIN$(K, \mathbf{A})$
2:     Compute similarity matrix $\mathbf{U}$ according to Eq.(3).
3:     Obtain $\mathcal{G} = \{G_1, \ldots, G_K\}$ with DIVIDE$(K, \mathbf{U})$.
4:     Pick out bad clients into set $\mathcal{B}$.
5:     Compute $\boldsymbol{\alpha}_{G_k} = \frac{1}{|G_k \backslash \mathcal{B}|} \sum_{C_i \in G_k \backslash \mathcal{B}} \boldsymbol{\alpha}_i^\star$ for $G_k$.
6:     Train model $\hat{h}_{\boldsymbol{\alpha}_{G_k}}$ by minimizing $\hat{\mathcal{L}}_{\boldsymbol{\alpha}_{G_k}}(h)$ and return $\hat{h}_{\boldsymbol{\alpha}_{G_k}}$ to all clients $C_i$ in group $G_k$.
7: **end procedure**
8: **function** DIVIDE$(K, \mathbf{U})$
9:     Pick $t$ random hyperplanes and use projection to divide the nodes into $2^t$ groups.
10:     Take the better solution of $t = 2$ and $t = 3$ with respect to the SDP relaxation Eq.(7).
11:     **return** Group partition $\mathcal{G} = \{G_1, \ldots, G_K\}$.
12: **end function**

---

groups $K$. If each group $G_k$ is not too small, *i.e.*, $N_k \geqslant 2\sqrt{Z_{in}}, \forall k \in \{1, \ldots, K\}$, then the number of bad clients $|\mathcal{B}|$ is upper bounded by the number of weak edges.

We further theoretically analyze the generalization bound of the models returned for the clients. Given group partition $\mathcal{G} = \{G_1, \ldots, G_K\}$ returned by solving the SDP, we remove all the weak edges within all groups. For any good clients $C_i$ and $C_j$ in the same group, there always exits a shortest path $p_{ij}$ between them since good clients in the same group are connected through strong edges. Let $\eta$ denote the maximum $p_{ij}$ between any good clients $C_i$ and $C_j$ in group $G_k, \forall k \in \{1, \ldots, K\}$. Denote $\tau = \min_{e_{ij} \in E_{in}} w_{ij}$ where $E_{in}$ is the set of strong edges within all groups. We provide Theorem 3 to show that the model returned by Algorithm 1 for each good client is an approximation of its *personalized* model.

**Theorem 3.** *Let $\mathcal{G} = \{G_1, \ldots, G_K\}$ be the group partition returned by solving the SDP. $\hat{h}_{\boldsymbol{\alpha}_{G_k}}$ is the model returned by Algorithm 1 for client $C_i$ in group $G_k$. Let $\mathrm{upp}(\hat{h}_{\boldsymbol{\alpha}_{G_k}})$ be the upper bound of the expected risk of $\hat{h}_{\boldsymbol{\alpha}_{G_k}}$ on $C_i$ according to the right-hand side of Eq.(1) and $\mathrm{upp}(\hat{h}_{\boldsymbol{\alpha}_i^\star})$ is the upper bound of the expected risk of the personalized model $\hat{h}_{\boldsymbol{\alpha}_i^\star}$ on $C_i$ according to the right-hand side of Eq.(1). The following result holds except for the bad clients in $\mathcal{B}$:*

$$\mathrm{upp}(\hat{h}_{\boldsymbol{\alpha}_{G_k}}) - \mathrm{upp}(\hat{h}_{\boldsymbol{\alpha}_i^\star}) \leqslant O\left( \eta(1 - \tau) \sqrt{\frac{N}{N - 1}} \right).$$

We leave the proof of Theorem 3 in Appendix C due to space limitation. Theorem 3 reveals that the generalization gap between the model returned by Algorithm 1 for any good client and its *personalized* model relies on the minimum weight of strong edges $\tau$ within all groups and the maximum length of the shortest paths between good clients in the same group $\eta$. These are further determined by the intrinsic properties of $\mathbf{U}$ with respect to modularity and the number of groups $K$.

### 4.2 Collaboration with Clustering

Suppose we have the prior knowledge about the collaboration partnerships among the clients, *i.e.*, there exists a potential partition $\mathcal{P}^\star = \{P_1^\star, \ldots, P_K^\star\}$. In this case, we try to develop the method with tighter bound for the approximated models. We use $\Phi(\mathcal{P})$ to evaluate the quality of partitions which is defined as $\Phi(\mathcal{P}) = \sum_{k=1}^{K} \sum_{C_i \in P_k} d(\boldsymbol{\alpha}_i^\star, \bar{\boldsymbol{\alpha}}_k)$, where $\mathcal{P} = \{P_1, \ldots, P_K\}$ is the group partition, $d(\cdot, \cdot)$ is the distance measure and $\bar{\boldsymbol{\alpha}}_k$ is the center of group $P_k$. If a partition $\mathcal{P}$ is of high quality, then $\Phi(\mathcal{P})$ is small. Take Figure 2 for example where $K = 2$, the value $\Phi(\mathcal{P})$ of the graph $A$ in Figure 2 is smaller than that in Figure 1 since clients in the same group in Figure 2 have more similar incoming edges. Thus the example in Figure 2 has better structures than that in Figure 1.

Let $\mathrm{OPT}_{\Phi(\mathcal{P})} = \min_{\mathcal{P}} \Phi(\mathcal{P})$ be the minimum value of $\Phi(\mathcal{P})$ over all group partitions and $\mathcal{P}^\star = \{P_1^\star, \ldots, P_K^\star\}$ is the corresponding optimal group partition. The distance between two group partitions $\mathcal{P} = \{P_1, \ldots, P_K\}$ and $\mathcal{P}' = \{P_1', \ldots, P_K'\}$ is denoted as $dist(\mathcal{P}, \mathcal{P}') = \min_{\sigma \in \mathfrak{S}_K} \frac{1}{N} \sum_{k=1}^{K} \mathbb{I}(P_k = P_{\sigma(k)}')$, where $\mathfrak{S}_K$ is the set of permutations. Balcan et al. (2009) proposed the $(1 + \gamma, \epsilon)$-*approximation-stability* property to show that if a given group partition

$\mathcal{P}$ satisfies $\Phi(\mathcal{P}) \leqslant (1 + \gamma)\text{OPT}_{\Phi(\mathcal{P})}$, then we have $dist(\mathcal{P}, \mathcal{P}^\star) \leqslant \epsilon$. This provides a way to find good group partitions by optimizing $\Phi(\mathcal{P})$. We assume that $\{\boldsymbol{\alpha}_1^\star, \dots, \boldsymbol{\alpha}_N^\star\}$ satisfy the $(1 + \gamma, \epsilon)$-approximation-stability property with respect to $\Phi(\mathcal{P})$.

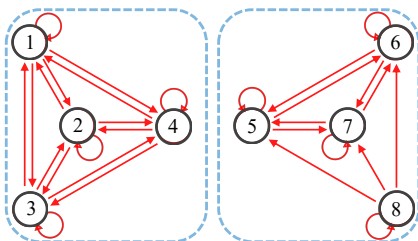

Figure 2: An example of the directed graph $A$ with potential structures.

In this case, there still exist some bad clients and we need to find them out. Let $d_1(\boldsymbol{\alpha}_i^\star) = \min_k d(\boldsymbol{\alpha}_i^\star, \bar{\boldsymbol{\alpha}}_k^\star)$ be the distance of $\boldsymbol{\alpha}_i^\star$ to its closest group center and $d_2(\boldsymbol{\alpha}_i^\star) = \min_{j \neq k} d(\boldsymbol{\alpha}_i^\star, \bar{\boldsymbol{\alpha}}_j^\star)$ is the distance of $\boldsymbol{\alpha}_i^\star$ to its second-closest group center. Denote $\bar{d} = \frac{1}{N}\text{OPT}_{\Phi(\mathcal{P})}$ as the average distance of all clients. Given any constant $t$ where $t > 2$, let $d^\star = \frac{\gamma\bar{d}}{\epsilon t}$ be the *critical distance*. We call $C_i$ the *bad client* if $d_1(\boldsymbol{\alpha}_i^\star) \geqslant d^\star$ or $d_2(\boldsymbol{\alpha}_i^\star) - d_1(\boldsymbol{\alpha}_i^\star) \leqslant \frac{t}{2}d^\star$, else we call it the *good client*. It is easy to understand that good clients are those tightly coupled to clients in the same group and loosely coupled to clients in different groups.

To guarantee the performance of the learned models, we pick out the bad clients. We return $\hat{h}_{\boldsymbol{\alpha}_{P_k}}$ for clients in group $P_k$ which is obtained by minimizing $\hat{\mathcal{L}}_{\boldsymbol{\alpha}_{P_k}}(h)$ where $\boldsymbol{\alpha}_{P_k} = \frac{1}{|P_k \setminus \mathcal{B}|} \sum_{C_i \in P_k \setminus \mathcal{B}} \boldsymbol{\alpha}_i^\star$. We refer this method as **A**daptive **C**ollaborative **L**earning with **C**lustering (ACLC). The whole process is summarized in Algorithm 2, where $\beta > 1$ and $t > 2$ are given constants. We use the method in Balcan et al. (2009) to obtain the group partition, where the constant-factor $K$-median approximation algorithm is done as that in Arya et al. (2004). Let $\mathcal{B}$ be the set of bad clients. We provide the following Lemma 3 to bound the number of bad clients $|\mathcal{B}|$ in the group partition $\mathcal{P}$ produced by Algorithm 2.

**Lemma 3.** *Assume that $\{\boldsymbol{\alpha}_1^\star, \dots, \boldsymbol{\alpha}_N^\star\}$ satisfy the $(1 + \gamma, \epsilon)$-approximation-stability property. Let $\mathcal{P} = \{P_1, \dots, P_K\}$ be the group partition produced by Algorithm 2. Then the number of bad clients $|\mathcal{B}|$ satisfies $|\mathcal{B}| < (6 + \frac{t}{\gamma})\beta\epsilon N$, where $t > 2$ and $\beta > 1$ are given constants.*

The proof of Lemma 3 can be found in Appendix D. Lemma 3 reveals that the number of bad clients is related to $\gamma$ and $\epsilon$, which reveals the degree of difficulty in finding approximate group partitions. We further theoretically analyze the generalization bound of the models returned for the clients in the following Theorem 4, which shows that the model returned by Algorithm 2 for each good client is an approximation of its *personalized* model.

**Theorem 4.** *Assume that $\{\boldsymbol{\alpha}_1^\star, \dots, \boldsymbol{\alpha}_N^\star\}$ satisfy the $(1 + \gamma, \epsilon)$-approximation-stability property. Let $\mathcal{P} = \{P_1, \dots, P_K\}$ be the group partition produced by Algorithm 2. $\hat{h}_{\boldsymbol{\alpha}_{P_k}}$ is the model returned by Algorithm 2 for client $C_i$ in group $P_k$. Let $\text{upp}(\hat{h}_{\boldsymbol{\alpha}_{P_k}})$ be the upper bound of the expected risk of $\hat{h}_{\boldsymbol{\alpha}_{P_k}}$ on $C_i$ according to the right-hand side of Eq.(1) and $\text{upp}(\hat{h}_{\boldsymbol{\alpha}_i^\star})$ is the upper bound of the expected risk of the personalized model $\hat{h}_{\boldsymbol{\alpha}_i^\star}$ on $C_i$ according to the right-hand side of Eq.(1). Then given any constants $t > 2$ and $\beta > 1$, the following result holds except for the bad clients in $\mathcal{B}$:*

$$upp(\hat{h}_{\boldsymbol{\alpha}_{P_k}}) - upp(\hat{h}_{\boldsymbol{\alpha}_i^\star}) \leqslant O\left(\frac{\gamma\text{OPT}_{\Phi(\mathcal{P})}}{\epsilon t N}\right).$$

We leave the proof of Theorem 4 in Appendix D due to space limitation. Theorem 4 reveals that the generalization gap between the model returned by Algorithm 2 for any good client and its personalized model relies on the potential structures of the clients' collaboration relationships, *i.e.*, the optimal value of objective $\Phi(\mathcal{P})$. If the collaboration relationships have good structures, *i.e.*, $\text{OPT}_{\Phi(\mathcal{P})}$ is small, then the returned model for any good client is close to its personalized model.

**Algorithm 2** Adaptive Collaborative Learning with Clustering (ACLC)

1:  **procedure** TRAIN($K$, $\mathbf{A}$, constants $\epsilon, \gamma, t, \beta$)
2:      Obtain $\mathcal{P} = \{P_1, \ldots, P_K\}$ and $\hat{d}$ with DIVIDE($K$, $\mathbf{A}$, $\epsilon, \gamma, t, \beta$).
3:      Compute the critical distance $\hat{d}^\star = \frac{\gamma \hat{d}}{\beta t \epsilon}$. Pick out bad clients into set $\mathcal{B}$.
4:      Compute $\boldsymbol{\alpha}_{P_k} = \frac{1}{|P_k \setminus \mathcal{B}|} \sum_{C_i \in P_k \setminus \mathcal{B}} \boldsymbol{\alpha}_i^\star$ for $P_k$.
5:      Train model $\hat{h}_{\boldsymbol{\alpha}_{P_k}}$ by minimizing $\hat{\mathcal{L}}_{\boldsymbol{\alpha}_{P_k}}(h)$ and return $\hat{h}_{\boldsymbol{\alpha}_{P_k}}$ to all clients $C_i$ in group $P_k$.
6:  **end procedure**
7:  **function** DIVIDE($K$, $\mathbf{A}$, $\epsilon, \gamma, t, \beta$)
8:      Run the constant-factor $K$-median approximation algorithm on $\{\boldsymbol{\alpha}_1^\star, \ldots, \boldsymbol{\alpha}_N^\star\}$ to compute $\Phi(\mathcal{P})$.
9:      Compute the value $\hat{d} = \frac{\Phi(\mathcal{P})}{N} \leqslant \beta \bar{d}$.
10:     Construct the threshold graph $G$ which connects all pairs $(C_i, C_j)$ with $d(\boldsymbol{\alpha}_i^\star, \boldsymbol{\alpha}_j^\star) < \frac{2\gamma \hat{d}}{\beta t \epsilon}$.
11:     **for** $k = 1$ to $K - 1$ **do**
12:         Let $v_i$ be the node with the highest degree in $G$.
13:         Select $v_i$ and its neighborhoods to get group $P_k$. Remove $P_k$ from $G$.
14:     **end for**
15:     **return** $\mathcal{P} = \{P_1, \ldots, P_K\}$ and $\hat{d}$, where $P_K = \{\boldsymbol{\alpha}_1^\star, \ldots, \boldsymbol{\alpha}_N^\star\} \setminus \left( \cup_{k=1}^{K-1} P_k \right)$.
16: **end function**

## 5  Experiments

In the experiments, we use the following four datasets: (1) The MNIST dataset consists of 70K handwritten digit images in 10 classes, which has a training set of 60K examples and a test set of 10K examples. (2) The CIFAR-10/100 dataset consists of 60K color images in $10/100$ classes, with $6000/600$ images per class. They have 50K training images and 10K test images. (3) The Federated Extended MNIST (FEMNIST) dataset is built by partitioning the data in Extended MNIST based on the writer of the digit/character. For MNIST, we use the same CNN model as that in McMahan et al. (2017); for CIFAR-10, we use the CNN model which consists of 2 convolutional layers followed by 2 fully connected layers as that in Ghosh et al. (2020); for CIFAR-100, we use the ResNet-18 model (He et al., 2016); for FEMNIST, we use the same CNN model as that in LEAF (Caldas et al., 2018). In collaborative learning, using KL divergence to estimate the divergence between distributions is a common practice (He et al., 2020; Yang, 2021). We use the right hand of Eq. 8 to estimate the IPM $d_{\mathcal{H}}(\mathcal{D}_i, \mathcal{D}_j)$ in the experiments since

$$d_{\mathcal{H}}(\mathcal{D}_i, \mathcal{D}_j) \leqslant \min \left\{ 1 - \frac{1}{2} e^{-d_{\mathrm{JS}}(\mathcal{D}_i \| \mathcal{D}_j)}, \sqrt{\frac{d_{\mathrm{JS}}(\mathcal{D}_i \| \mathcal{D}_j)}{2}} \right\}, \tag{8}$$

where $d_{\mathrm{JS}}(\mathcal{D}_i \| \mathcal{D}_j)$ is the Jensen-Shannon (JS) divergence (symmetrical KL divergence). The proof can be found in Appendix E.

First, we partition MNIST into clients using the same method as that in McMahan et al. (2017) and set client number $N = 10$ where each client has 3 classes, total number of samples $M = 2000$. We partition CIFAR-10/100 using the Dirichlet partition as that in Yurochkin et al. (2019) and Wang et al. (2020a). We set $N = 20$ and $M = 5000$ for CIFAR-10 and set $N = 50$ and $M = 20000$ for CIFAR-100. For FEMNIST, we use the default partition proposed in LEAF and choose $N = 20$ clients randomly from the total 3550 clients (Caldas et al., 2018). We further set the number of groups $K = 3$ for MNIST, $K = 4$ for CIFAR-10, $K = 7$ for CIFAR-100 and $K = 4$ for FEMNIST.

We divide $N$ clients into $K$ groups and return a model for each group according to Algorithm 1 on the three datasets respectively, which is denoted as ACLMM. We use the *centralized* model as the baseline, where only one single model is trained according to $\alpha_{ij} = \frac{m_j}{M}, \forall i \in \{1, \ldots, N\}$ as that in McMahan et al. (2017). We also implement the *personalized* model, where each client will obtain the personalized model according to their $\boldsymbol{\alpha}_i^\star$. Since the client number on CIFAR-100 is relatively large, we only perform the *personalized* model on MNIST, CIFAR-10 and FEMNIST. The results on four datasets are depicted in Figure 3. From Figure 3, we find that the model learned with our ACLMM performs better than the centralized model and is comparable to the personalized model.

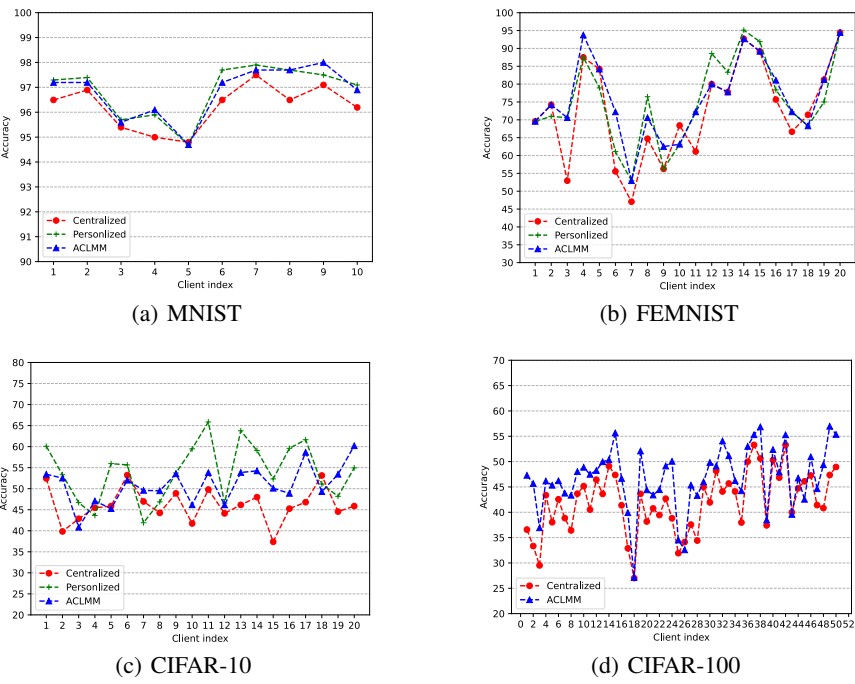

Figure 3: Comparisons of ACLMM with *Personalized* and *Centralized* models.

In order to verify the collaboration partners detected by our ACLMM, we take CIFAR-10 for example and visualize the group partition returned by ACLMM in Figure 4(a), where nodes with the same color are in the same group and thicker edge means larger similarity between clients. We also visualize the similarity matrix $\mathbf{U}$ as a color map in Figure 4(b), where the $(i, j)$-th block characterizes the similarity between $C_i$ and $C_j$ (the lighter the color is, the smaller the similarity is). Obviously, clients in the same group (having the same node color) in Figure 4(a) are those with large similarities in Figure 4(b), which means they are collaboration partners.

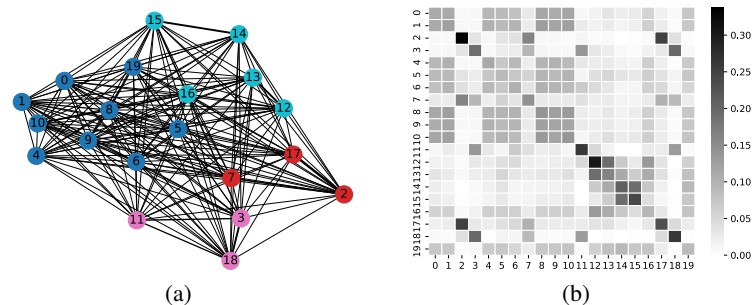

Figure 4: (a) The group partition of CIFAR-10 based on ACLMM. (b) The color map associated to $\mathbf{U}$ of CIFAR-10.

Next, we construct the clients that have intrinsic structures on MNIST and CIFAR-10/100 in the following way. For MNIST, we set $N = 10$ with $K = 3$. The first group contains 3 clients with samples dominated by digit $\{0, 1, 2\}$. The second group contains 3 clients with samples dominated by digit $\{3, 4, 5\}$. The third group contains 4 clients with samples dominated by digit $\{6, 7, 8, 9\}$. The number of samples on the clients satisfies the Dirichlet distribution as that in Yurochkin et al. (2019), and Wang et al. (2020a). We analogously set $N = 20$ with $K = 4$ for CIFAR-10 and set $N = 50$ with $K = 7$ for CIFAR-100. We set the total number of sample $M = 2000/5000/20000$ for MNIST/CIFAR-10/CIFAR-100 respectively. Since the partition of FEMNIST have ambiguous clustering structures (Ghosh et al., 2020), we did not run clustering experiments on FEMNIST.

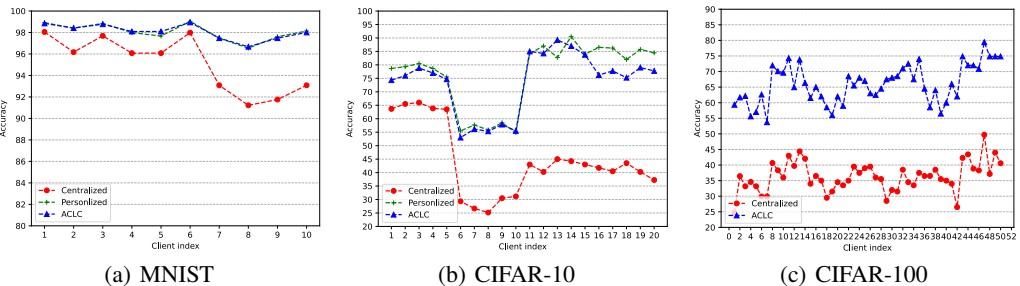

|  (a) MNIST  |  (b) CIFAR-10  |  (c) CIFAR-100  |

Figure 5: Comparisons of ACLC with *Personalized* and *Centralized* models.

We divide $N$ clients into $K$ groups and return a model for each group according to Algorithm 2 on the three datasets respectively, which is denoted as ACLC. We also implement the *centralized* model and the *personalized* model. We only perform the *personalized* model on MNIST and CIFAR-10. The results on three datasets are depicted in Figure 5. From Figure 5 we can find that the model learned with our ACLC performs much better than the centralized model and is comparable to the personalized model. Furthermore, the gap between the model return by ACLC and the personalized model is small, which shows that our bound in Theorem 4 is tight when the collaboration relationships have intrinsic clustering structures.

# 6   Conclusion

In this paper, we propose the adaptive collaborative learning method to deal with heterogeneous data on the clients, which can detect the collaboration partners from numerous heterogeneous clients and adaptively learn $K$ models for them. We also theoretically prove that the model learned with our method for each client is a good approximation of its personalized model. Experimental results on real-world datasets verify the effectiveness of our method.

## Broader Impact

Our work provides the possibility of lowering computational burden while maintaining comparable model performance in collaborative learning. We believe our work will be beneficial for collaborative applications, and do not have any negative societal impacts.

## Acknowledgment

We would like to thank Chiyu Cai for helpful discussions and anonymous reviewers for their valuable comments. This work is supported by the National Key Research and Development Program of China (2018AAA0101100), the National Science Foundation of China (61921006, 62276125), the Fundamental Research Funds for the Central Universities (022114380013), the project of HUAWEI-LAMDA Joint Laboratory of Artificial Intelligence, and the Collaborative Innovation Center of Novel Software Technology and Industrialization.

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
