# A  Proof of Theorem 1

*Proof.* We mainly use Hoeffding's inequality to prove Theorem 1. Notice that the Integral Probability Metrics (IPM) is defined as $d_{\mathcal{H}}(\mathcal{D}_i, \mathcal{D}_j) = \sup_{h \in \mathcal{H}} \left| \mathcal{L}_{\mathcal{D}_i}(h) - \mathcal{L}_{\mathcal{D}_j}(h) \right|$. For $\forall h \in \mathcal{H}$ and client $C_i, \forall i \in \{1, \ldots, N\}$, we have

$$
\left| \mathcal{L}_{\mathcal{D}_i}(h) - \hat{\mathcal{L}}_{\boldsymbol{\alpha}_i}(h) \right| = \left| \mathcal{L}_{\mathcal{D}_i}(h) - \sum_{j=1}^{N} \alpha_{ij} \hat{\mathcal{L}}_{S_j}(h) \right|
$$

$$
= \left| \sum_{j=1}^{N} \alpha_{ij} \mathcal{L}_{\mathcal{D}_i}(h) - \sum_{j=1}^{N} \alpha_{ij} \mathcal{L}_{\mathcal{D}_j}(h) + \sum_{j=1}^{N} \alpha_{ij} \mathcal{L}_{\mathcal{D}_j}(h) - \sum_{j=1}^{N} \alpha_{ij} \hat{\mathcal{L}}_{S_j}(h) \right|
$$

$$
= \left| \sum_{j=1}^{N} \alpha_{ij} \mathcal{L}_{\mathcal{D}_i}(h) - \sum_{j=1}^{N} \alpha_{ij} \mathcal{L}_{\mathcal{D}_j}(h) \right| + \left| \sum_{j=1}^{N} \alpha_{ij} \mathcal{L}_{\mathcal{D}_j}(h) - \sum_{j=1}^{N} \alpha_{ij} \hat{\mathcal{L}}_{S_j}(h) \right|
$$

$$
\leqslant \sum_{j=1}^{N} \alpha_{ij} \left| \mathcal{L}_{\mathcal{D}_i}(h) - \mathcal{L}_{\mathcal{D}_j}(h) \right| + \left| \sum_{j=1}^{N} \alpha_{ij} \mathcal{L}_{\mathcal{D}_j}(h) - \sum_{j=1}^{N} \alpha_{ij} \hat{\mathcal{L}}_{S_j}(h) \right|
$$

$$
\leqslant \sum_{j=1}^{N} \alpha_{ij} d_{\mathcal{H}}(\mathcal{D}_i, \mathcal{D}_j) + \left| \sum_{j=1}^{N} \alpha_{ij} \mathcal{L}_{\mathcal{D}_j}(h) - \sum_{j=1}^{N} \alpha_{ij} \hat{\mathcal{L}}_{S_j}(h) \right|.
$$

For the loss function $l$, let $\{X_1, \ldots, X_{m_1}\}$ be the random variables which take on values $\frac{\alpha_{i1} M}{m_1} l(h(\boldsymbol{x}), y)$ for the $m_1$ examples $(\boldsymbol{x}, y) \in S_1$ with respect to $h \in \mathcal{H}$. Random variables $\{X_{m_1+1}, \ldots, X_M\}$ are defined analogously. Then the weighted empirical risk $\sum_{j=1}^{N} \alpha_{ij} \hat{\mathcal{L}}_{S_j}(h)$ can be written as follows:

$$
\sum_{j=1}^{N} \alpha_{ij} \hat{\mathcal{L}}_{S_j}(h) = \sum_{t=1}^{N} \frac{\alpha_{ij}}{m_t} \sum_{j=1}^{m_t} l(h(\boldsymbol{x}_j^t), y_j^t) = \frac{1}{M} \sum_{j=1}^{M} X_j.
$$

By the linearity of expectations, we have

$$
\mathbb{E} \left[ \sum_{j=1}^{N} \alpha_{ij} \hat{\mathcal{L}}_{S_j}(h) \right] = \sum_{j=1}^{N} \alpha_{ij} \mathcal{L}_{\mathcal{D}_j}(h).
$$

Then the following result holds for every $h \in \mathcal{H}$ according to Hoeffding's inequality:

$$
\Pr \left[ \left| \sum_{j=1}^{N} \alpha_{ij} \hat{\mathcal{L}}_{S_j}(h) - \sum_{j=1}^{N} \alpha_{ij} \mathcal{L}_{\mathcal{D}_j}(h) \right| \geqslant \epsilon \right] \leqslant 2 \exp \left( -\frac{2 M^2 \epsilon^2}{\sum_{j=1}^{M} \text{Range}^2(X_j)} \right) = 2 \exp \left( -\frac{2 \epsilon^2}{\mu^2 \sum_{j=1}^{N} \frac{\alpha_{ij}^2}{m_j}} \right).
$$

By the definition of growth function $\Pi_{\mathcal{H}}(\cdot)$ and according to union bound, the following result holds for $\forall h \in \mathcal{H}$:

$$
\Pr \left[ \left| \sum_{j=1}^{N} \alpha_{ij} \hat{\mathcal{L}}_{S_j}(h) - \sum_{j=1}^{N} \alpha_{ij} \mathcal{L}_{\mathcal{D}_j}(h) \right| \geqslant \epsilon \right] \leqslant 4 \Pi_{\mathcal{H}}(2M) \exp \left( \frac{-2(\frac{1}{4}\epsilon)^2}{\mu^2 \sum_{j=1}^{N} \frac{\alpha_{ij}^2}{m_j}} \right)
$$

$$
\leqslant 4(2M)^d \exp \left( \frac{-\frac{1}{8}\epsilon^2}{\mu^2 \sum_{j=1}^{N} \frac{\alpha_{ij}^2}{m_j}} \right).
$$

Substituting $\delta$ for the probability gives the following result:

$$
\epsilon = \mu \sqrt{\sum_{j=1}^{N} \frac{\alpha_{ij}^2}{m_j}} \sqrt{8(d \log(2M) + \log \frac{4}{\delta})}.
$$

Note that $h_i^\star = \arg\min\limits_{h \in \mathcal{H}} \mathcal{L}_{\mathcal{D}_i}(h)$ and $\hat{h}_{\boldsymbol{\alpha}_i} = \arg\min\limits_{h \in \mathcal{H}} \hat{\mathcal{L}}_{\boldsymbol{\alpha}_i}(h)$. Given any $\delta \in (0, 1)$, the following result holds with probability at least $1 - \delta$:

$$\mathcal{L}_{\mathcal{D}_i}(\hat{h}_{\boldsymbol{\alpha}_i}) \leqslant \hat{\mathcal{L}}_{\boldsymbol{\alpha}_i}(\hat{h}_{\boldsymbol{\alpha}_i}) + \mu \sqrt{\sum_{j=1}^{N} \frac{\alpha_{ij}^2}{m_j}} \sqrt{8(d\log(2M) + \log\frac{8}{\delta})} + \sum_{j=1}^{N} \alpha_{ij} d_{\mathcal{H}}(\mathcal{D}_i, \mathcal{D}_j)$$

$$\leqslant \hat{\mathcal{L}}_{\boldsymbol{\alpha}_i}(h_i^\star) + \mu \sqrt{\sum_{j=1}^{N} \frac{\alpha_{ij}^2}{m_j}} \sqrt{8(d\log(2M) + \log\frac{8}{\delta})} + \sum_{j=1}^{N} \alpha_{ij} d_{\mathcal{H}}(\mathcal{D}_i, \mathcal{D}_j)$$

$$\leqslant \mathcal{L}_{\mathcal{D}_i}(h_i^\star) + 2\mu \sqrt{\sum_{j=1}^{N} \frac{\alpha_{ij}^2}{m_j}} \sqrt{8(d\log(2M) + \log\frac{8}{\delta})} + 2\sum_{j=1}^{N} \alpha_{ij} d_{\mathcal{H}}(\mathcal{D}_i, \mathcal{D}_j).$$

$\square$

## B  Proof of Theorem 2

*Proof.* The learning bound in Theorem 1 suggests minimizing the following objective with respect to $\boldsymbol{\alpha}_i$ for client $C_i$.

$$\min_{\boldsymbol{\alpha}_i} \quad \lambda \sqrt{\sum_{j=1}^{N} \frac{\alpha_{ij}^2}{m_j}} + \sum_{j=1}^{N} \alpha_{ij} d_{\mathcal{H}}(\mathcal{D}_i, \mathcal{D}_j) \tag{9}$$

$$s.t. \quad \alpha_{ij} \geqslant 0, \ \forall j \in \{1, \ldots, N\}, \ \textstyle\sum_{j=1}^{N} \alpha_{ij} = 1.$$

where $\lambda = \mu\sqrt{8(d\log(2M) + \log\frac{8}{\delta})}$. The Lagrangian function of Eq.(9) is

$$\mathbb{L}(\boldsymbol{\alpha}_i, \boldsymbol{\eta}, \zeta) = \lambda \sqrt{\sum_{j=1}^{N} \frac{\alpha_{ij}^2}{m_j}} + \sum_{j=1}^{N} \alpha_{ij} d_{\mathcal{H}}(\mathcal{D}_i, \mathcal{D}_j) - \boldsymbol{\alpha}_i^\top \boldsymbol{\eta} - \zeta(\mathbf{1}^\top \boldsymbol{\alpha}_i - 1).$$

To minimize the objective, the following Karush-Kuhn-Tucker (KKT) condition holds:

$$\begin{cases} \partial_{\boldsymbol{\alpha}_i} \mathbb{L}(\boldsymbol{\alpha}_i, \boldsymbol{\eta}, \zeta) = 0. \\ \boldsymbol{\alpha}_i \geqslant 0, \ \boldsymbol{\eta} \geqslant 0, \ \alpha_{ij}\eta_j = 0, \ \forall j \in \{1, \ldots, N\}. \\ \mathbf{1}^\top \boldsymbol{\alpha}_i = 1. \end{cases}$$

Let the partial derivative equals to zero with respect to $\forall t \in \{1, \ldots, N\}$:

$$\partial_{\boldsymbol{\alpha}_{it}} \mathbb{L}(\boldsymbol{\alpha}_i, \boldsymbol{\eta}, \zeta) = d_{\mathcal{H}}(\mathcal{D}_i, \mathcal{D}_t) - \eta_t - \zeta - \lambda \frac{\alpha_{it}}{m_t \sqrt{\sum_{j=1}^{N} \frac{\alpha_{ij}^2}{m_j}}} = 0.$$

Since $\alpha_{ij}\eta_j = 0, \ \forall j \in \{1, \ldots, N\}$, we discuss the following two cases:

(1) If $\alpha_{it} = 0$, then $\eta_t = d_{\mathcal{H}}(\mathcal{D}_i, \mathcal{D}_t) - \zeta \geqslant 0$;

(2) If $\alpha_{it} > 0$, then $\eta_t = 0$. In this case,

$$\alpha_{it} = \frac{m_t \sqrt{\sum_{j=1}^{N} \frac{\alpha_{ij}^2}{m_j}}[\zeta - d_{\mathcal{H}}(\mathcal{D}_i, \mathcal{D}_t)]}{\lambda} > 0.$$

Denote $\mathcal{Q}_i = \{t \mid \alpha_{it} > 0\}$. Notice that

$$\zeta - d_{\mathcal{H}}(\mathcal{D}_i, \mathcal{D}_q) \begin{cases} > 0, & q \in \mathcal{Q}_i, \\ \leqslant 0, & q \notin \mathcal{Q}_i. \end{cases} \tag{10}$$

Thus we sort the clients according to $d_{\mathcal{H}}(\mathcal{D}_i, \mathcal{D}_j)$. For convenience, we denote $\Xi_i^j = d_{\mathcal{H}}(\mathcal{D}_i, \mathcal{D}_j)$ for client $C_i$ where $j \in \{1, \ldots, N\}$, $\forall i \in \{1, \ldots, N\}$. Sort $\{\Xi_i^1, \ldots, \Xi_i^N\}$ in ascending order to get $\{\Xi_i^{\sigma(1)}, \ldots, \Xi_i^{\sigma(N)}\}$, $i.e.$, $\Xi_i^{\sigma(1)} \leqslant \ldots \leqslant \Xi_i^{\sigma(N)}$, where $\sigma(\cdot) : [N] \to [N]$ is a bijection which represents the initial index.

Notice that

$$\sum_{q \in \mathcal{Q}_i} \frac{\alpha_{iq}^2}{m_q} = \frac{\sum_{q \in \mathcal{Q}_i} m_q \sum_{j=1}^N \frac{\alpha_{ij}^2}{m_j} [\zeta - d_{\mathcal{H}}(\mathcal{D}_i, \mathcal{D}_q)]^2}{\lambda^2},$$

and for indexes $q \in \mathcal{Q}_i$,

$$\sum_{q \in \mathcal{Q}_i} \frac{\alpha_{iq}^2}{m_q} = \sum_{j=1}^N \frac{\alpha_{ij}^2}{m_j}.$$

Thus we get

$$\sum_{q \in \mathcal{Q}_i} m_q [\zeta - d_{\mathcal{H}}(\mathcal{D}_i, \mathcal{D}_q)]^2 = \lambda^2. \tag{11}$$

The discriminant of Eq.(11) should satisfy the following property:

$$\left( \sum_{q \leqslant q_i} m_{\sigma(q)} \Xi_i^{\sigma(q)} \right)^2 - \left( \sum_{q \leqslant q_i} m_{\sigma(q)} \right) \left( \sum_{q \leqslant q_i} m_{\sigma(q)} (\Xi_i^{\sigma(q)})^2 - \lambda^2 \right) \geqslant 0, \tag{12}$$

where $q_i$ is the largest index that makes Eq.(12) hold. Thus $\zeta$ is the larger solution of Eq.(11). In addition, $\zeta$ should satisfies Eq.(10). Thus

$$q_i = \arg\max_t \left\{ t \,\Big|\, \zeta \geqslant \Xi_i^{\sigma(t)} \wedge \left( \sum_{q \leqslant t} m_{\sigma(q)} \Xi_i^{\sigma(q)} \right)^2 \geqslant \left( \sum_{q \leqslant t} m_{\sigma(q)} \right) \left( \sum_{q \leqslant t} m_{\sigma(q)} (\Xi_i^{\sigma(q)})^2 - \lambda^2 \right) \right\}.$$

Notice that $\mathbf{1}^\top \boldsymbol{\alpha}_i = 1$, thus we have

$$\sum_{q \in \mathcal{Q}_i} \alpha_q = \frac{\sum_{q \in \mathcal{Q}_i} m_q \sqrt{\sum_{j=1}^N \frac{\alpha_{ij}^2}{m_j}} [\zeta - d_{\mathcal{H}}(\mathcal{D}_i, \mathcal{D}_q)]}{\lambda} = 1.$$

Thus we obtain

$$\sqrt{\sum_{j=1}^N \frac{\alpha_{ij}^2}{m_j}} = \frac{\lambda}{\sum_{q \in \mathcal{Q}_i} m_q [\zeta - d_{\mathcal{H}}(\mathcal{D}_i, \mathcal{D}_q)]}.$$

Thus we get the required result

$$\alpha_{ij}^\star = \left[ \frac{m_j (\zeta - \Xi_i^j)}{\sum_{q \leqslant q_i} m_{\sigma(q)} (\zeta - \Xi_i^{\sigma(q)})} \right]_+,$$

where $[\cdot]_+ = max(\cdot, 0)$.

$\square$

## C  Proof of Theorem 3 and Lemma 2

First we prove that maximizing Eq.(6) is equivalent to maximizing Eq.(7).

**Lemma 4.** *Maximizing the objective* $\sum_{i,j} (\frac{w_{ij}}{2W} - \frac{d_i d_j}{4W^2}) \boldsymbol{\nu}_i \cdot \boldsymbol{\nu}_j$ *is equivalent to maximizing objective* $\sum_{\mathcal{M}^+} \mathcal{M}_{ij} \boldsymbol{\nu}_i \cdot \boldsymbol{\nu}_j + \sum_{\mathcal{M}^-} -\mathcal{M}_{ij} (1 - \boldsymbol{\nu}_i \cdot \boldsymbol{\nu}_j)$ *where* $\mathcal{M} = \frac{w_{ij}}{2W} - \frac{d_i d_j}{4W^2}$.

*Proof.* Note that $\sum_{ij} \mathcal{M}_{ij} = 0$, thus $P = \sum_{\mathcal{M}^+} \mathcal{M}_{ij} = \sum_{\mathcal{M}^-} -\mathcal{M}_{ij}$ is a constant. Add constant $P$ to the original objective, then the new objective is

$$\sum_{i,j}(\frac{w_{ij}}{2W} - \frac{d_i d_j}{4W^2})\boldsymbol{\nu}_i \cdot \boldsymbol{\nu}_j + P = \sum_{i,j} \mathcal{M}_{ij}\boldsymbol{\nu}_i \cdot \boldsymbol{\nu}_j + \sum_{\mathcal{M}^-} -\mathcal{M}_{ij}$$

$$= \sum_{\mathcal{M}^+} \mathcal{M}_{ij}\boldsymbol{\nu}_i \cdot \boldsymbol{\nu}_j + \sum_{\mathcal{M}^-} -\mathcal{M}_{ij}\left(1 - \boldsymbol{\nu}_i \cdot \boldsymbol{\nu}_j\right).$$

Thus maximizing these two objective is equivalent. $\square$

Then we prove Lemma 2.

*Proof.* Suppose $\mathcal{G} = \{G_1, \ldots, G_K\}$ is the group partition returned by Algorithm 1. Let $Q(\mathcal{G})$ be the modularity of $\mathcal{G}$. We have $Q(\mathcal{G}) > \kappa \text{OPT}_{Q(\mathcal{G})} - (1 - \kappa)$ according to Lemma 1. Recall that modularity is defined as $Q(\mathcal{G}) = \frac{1}{W} \sum_k^K W_{in}^{G_k} - \frac{1}{4W^2} \sum_k^K (W_{vol}^{G_k})^2$. Note that $\sum_k^K W_{vol}^{G_k} = 2W$. By Cauchy inequality,

$$\frac{1}{4W^2} \sum_k^K (W_{vol}^{G_k})^2 \geqslant \frac{1}{4W^2} \frac{1}{K} (\sum_k^K W_{vol}^{G_k})^2 = \frac{1}{K}.$$

Thus we get

$$\sum_k^K W_{in}^{G_k} \geqslant W(Q(\mathcal{G}) + \frac{1}{K}) \geqslant W(\kappa \text{OPT}_{Q(\mathcal{G})} - (1 - \kappa) + \frac{1}{K}).$$

Since $\mathbf{U}$ is defined as $\mathbf{U} = \mathbf{D}_{in}^{-\frac{1}{2}} \mathbf{A}\mathbf{A}^T \mathbf{D}_{in}^{-\frac{1}{2}}$, the weight of *strong edge* in $\mathbf{U}$ satisfies $w_{ij} \leqslant 1$. According to the definition of *weak edge*, the weight of *weak edge* satisfies $w_{ij} \leqslant \frac{1}{N}$. Let $\mathcal{E}_{in}$ be the set of all intra-group edges. The total number of intra-group edges $|\mathcal{E}_{in}| = \sum_{k=1}^K \frac{N_k(N_k-1)}{2} = \frac{1}{2}\left(\sum_{k=1}^K N_k^2 - N\right)$. We have that

$$\sum_k^K W_{in}^{G_k} = \sum_{e_{ij} \in Z_{in}} w_{ij} + \sum_{e_{ij} \in \mathcal{E}_{in} \backslash Z_{in}} w_{ij}$$

$$\leqslant \sum_{e_{ij} \in Z_{in}} \frac{1}{N} + \sum_{e_{ij} \in \mathcal{E}_{in} \backslash Z_{in}} (|\mathcal{E}_{in}| - |Z_{in}|).$$

Thus we obtain

$$Z_{in} \leqslant \frac{N}{N-1}\left(|\mathcal{E}_{in}| - \sum_k^K W_{in}^{G_k}\right)$$

$$\leqslant \frac{N}{2(N-1)}\left[\sum_{k=1}^K N_k^2 - N - 2W\left(\kappa \text{OPT}_{Q(\mathcal{G})} - (1 - \kappa) + \frac{1}{K}\right)\right]$$

$$\leqslant \frac{N}{2(N-1)}\left[\frac{N^2}{K} - N - 2W\left(\kappa \text{OPT}_{Q(\mathcal{G})} + \kappa - 1 + \frac{1}{K}\right)\right]$$

$$= \frac{N}{2(N-1)}\left[\frac{N^2 - KN}{K} - 2W\left((\kappa + 1)\text{OPT}_{Q(\mathcal{G})} - \frac{K-1}{K}\right)\right].$$

Assume there are $x_k$ *bad client* in group $G_k$, which will yield $x_k(N_k - x_k)$ *weak edge* in group $G_k$. Thus we have

$$Z_{in} \geqslant \sum_{k=1}^K x_k(N_k - x_k) \geqslant x_1(N_{min} - x_1).$$

Since $N_{min} \geqslant \sqrt{2Z_{in}}$, we have $x_1 \leqslant \frac{N_{min}-\sqrt{N_{min}^2-4Z_{in}}}{2} \leqslant \frac{N_1}{2}$. Since $N_{min} = \min_k N_k$, one *bad client* in other groups will yield more *weak edge* than in $G_{min}$ corresponding to $N_{min}$. Thus $|\mathcal{B}| \leqslant \frac{N_{min}-\sqrt{N_{min}^2-4Z_{in}}}{2}$.

□

To prove Theorem 3, we first provide the following supporting lemma.

**Lemma 5.** *Suppose two vectors $\boldsymbol{\alpha}, \boldsymbol{\beta} \in \mathbb{R}^N$ which satisfy $\sum_{i=1}^N \alpha_i \beta_i \geqslant \tau$ where $\tau \geqslant \frac{1}{N}$ is a constant, then*

$$\sum_{i=1}^N (\alpha_i - \beta_i)^2 \leqslant (1-\tau)^2 \frac{N}{N-1}.$$

*Proof.* We seek to maximize the distance between $\boldsymbol{\alpha}$ and $\boldsymbol{\beta}$, which can be formalized as follows.

$$\begin{aligned}
\min_{\boldsymbol{\alpha},\boldsymbol{\beta}} \quad & -\|\boldsymbol{\alpha} - \boldsymbol{\beta}\|_2^2 \\
s.t. \quad & \textstyle\sum_{i=1}^N \alpha_i = 1, \ \sum_{i=1}^N \beta_i = 1, \\
& \textstyle\sum_{i=1}^N \alpha_i \beta_i \geqslant \tau.
\end{aligned} \tag{13}$$

The Lagrangian function of Eq.(13) is

$$\mathbb{L}(\boldsymbol{\alpha}, \boldsymbol{\beta}, \lambda_1, \lambda_2, \lambda_3) = -\sum_{i=1}^N (\alpha_i - \beta_i)^2 - \lambda_1 (\sum_{i=1}^N \alpha_i - 1) - \lambda_2 (\sum_{i=1}^N \beta_i - 1) - \lambda_3 (\sum_{i=1}^N \alpha_i \beta_i - \tau).$$

The following Karush-Kuhn-Tucker (KKT) condition holds.

$$\begin{cases}
\partial_{\boldsymbol{\alpha}} \mathbb{L}(\boldsymbol{\alpha}, \boldsymbol{\beta}, \lambda_1, \lambda_2, \lambda_3) = 0, \ \partial_{\boldsymbol{\beta}} \mathbb{L}(\boldsymbol{\alpha}, \boldsymbol{\beta}, \lambda_1, \lambda_2, \lambda_3) = 0, \\
\sum_{i=1}^N \alpha_i \beta_i \geqslant \tau, \ \lambda_3 \geqslant 0, \ \lambda_3 (\sum_{i=1}^N \alpha_i \beta_i - \tau) = 0, \\
\sum_{i=1}^N \alpha_i = 1, \ \sum_{i=1}^N \beta_i = 1.
\end{cases}$$

Let the partial derivative equals to zero with respect to $\alpha_i, \forall i \in \{1, \ldots, N\}$.

$$\partial_{\alpha_i} \mathbb{L}(\boldsymbol{\alpha}, \boldsymbol{\beta}, \lambda_1, \lambda_2, \lambda_3) = -2(\alpha_i - \beta_i) - \lambda_1 - \lambda_3 \beta_i = 0.$$

Thus we get

$$\alpha_i = \frac{(2 - \lambda_3)\beta_i - \lambda_1}{2}.$$

Note that $\sum_{i=1}^N \alpha_i = 1$ and $\sum_{i=1}^N \beta_i = 1$.

$$\sum_{i=1}^N \alpha_i = \sum_{i=1}^N \frac{(2 - \lambda_3)\beta_i - \lambda_1}{2} = 1 - \frac{\lambda_3}{2} - \frac{\lambda_1}{2} N = 1.$$

Thus $\lambda_3 = -N\lambda_1$. Analogously, $\lambda_3 = -N\lambda_2$. Set $\lambda_1 = \lambda_2 = -\lambda \neq 0$ and thus $\lambda_3 = N\lambda \neq 0$. Substituting $\lambda_1$ with $-\lambda$ and $\lambda_3$ with $N\lambda$, we have

$$\alpha_i = \frac{(2 - N\lambda)\beta_i + \lambda}{2}.$$

Since $\lambda_3(\sum_{i=1}^N \alpha_i \beta_i - \tau) = 0$, we have $\sum_{i=1}^N \alpha_i \beta_i = \tau$.

$$\sum_{i=1}^N \alpha_i \beta_i = \sum_{i=1}^N \frac{(2 - N\lambda)\beta_i^2 + \lambda \beta_i}{2} = \frac{\lambda}{2} + \frac{2 - N\lambda}{2} \sum_{i=1}^N \beta_i^2 = \tau.$$

Denote $\sum_{i=1}^N \beta_i^2 = x$, then

$$\lambda = \frac{2x - 2\tau}{Nx - 1} > 0.$$

Thus $x > \tau \geqslant \frac{1}{N}$. Since $1 \geqslant \sum_{i=1}^{N} \beta_i^2 \geqslant \frac{1}{N} \left( \sum_{i=1}^{N} \beta_i \right)^2 = \frac{1}{N}$, thus $x \in [\tau, 1]$.

The distance is then

$$\sum_{i=1}^{N} (\alpha_i - \beta_i)^2 = \sum_{i=1}^{N} (\frac{\lambda - N\lambda\beta_i}{2})^2 = \frac{N(x-\tau)^2}{(Nx-1)}.$$

Let $f(x) = \frac{(x-\tau)^2}{(Nx-1)}$ with its derivative

$$f'(x) = \frac{Nx^2 - 2x + 2\tau - N\tau^2}{(Nx-1)^2}.$$

Obviouly, $f'(x) \geqslant 0$ when $x \in [\tau, 1]$. Thus $f(x)$ is monotonically increasing for $x \in [\tau, 1]$. Thus we get the required result

$$\sum_{i=1}^{N} (\alpha_i - \beta_i)^2 \leqslant Nf(1) = (1-\tau)^2 \frac{N}{N-1}.$$

$\square$

We use the above lemma to prove Theorem 3.

*Proof.* Since $\mathbf{U}$ is defined as $\mathbf{U}_{in} = \mathbf{D}_{in}^{-\frac{1}{2}} \mathbf{A}\mathbf{A}^T \mathbf{D}_{in}^{-\frac{1}{2}}$, its elements are expressed as $U_{ij} = \frac{1}{\sqrt{d_i^{in} d_j^{in}}} \sum_{t=1}^{N} A_{it} A_{jt}$. Note that $d_i^{in} = \sum_{j=1}^{N} \alpha_{ij}^\star = 1$ and $A_{it} = \alpha_{it}^\star$. Thus $U_{ij} = \sum_{t=1}^{N} \alpha_{it}^\star \alpha_{jt}^\star$. Considering two good clients $C_i$ and $C_j$. If $e_{ij}$ is a strong edge, *i.e.*, $w_{ij} \geqslant \frac{1}{N}$, then $\|\boldsymbol{\alpha}_i^\star - \boldsymbol{\alpha}_j^\star\|_2 \leqslant (1-\tau)\sqrt{\frac{N}{N-1}}$ according to Lemma 5. Otherwise, $C_i$ can reach $C_j$ through a path whose length is less than $\eta$. According to the triangle inequality, $\|\boldsymbol{\alpha}_i^\star - \boldsymbol{\alpha}_j^\star\|_2 \leqslant \eta(1-\tau)\sqrt{\frac{N}{N-1}}$.

Note that $f(x) = x^2$ is a convex function on $[0, \infty)$. According to Jensen inequality, $f(\frac{1}{N} \sum_{i=1}^{N} |x_i|) \leqslant \frac{1}{N} \sum_{i=1}^{N} f(|x_i|)$. Thus

$$\left( \frac{1}{N} \sum_{i=1}^{N} |x_i| \right)^2 \leqslant \frac{1}{N} \sum_{i=1}^{N} |x_i|^2 \Rightarrow \|\boldsymbol{x}\|_1 = \sum_{i=1}^{N} |x_i| \leqslant \sqrt{N \sum_{i=1}^{N} |x_i|^2} = \sqrt{N}\|\boldsymbol{x}\|_2.$$

Thus we get the required result

$$\begin{aligned}
upp(\hat{h}_{G_k}) - upp(\hat{h}_{\boldsymbol{\alpha}_i^\star}) &= 2\sum_{j=1}^{N} \left( \alpha_{G_k j} - \alpha_{ij}^\star \right) d_{\mathcal{H}}(\mathcal{D}_i, \mathcal{D}_j) + 2\lambda \left( \sqrt{\sum_{j=1}^{N} \frac{\alpha_{G_k j}^2}{m_j}} - \sqrt{\sum_{j=1}^{N} \frac{\alpha_{ij}^{\star 2}}{m_j}} \right) \\
&\leqslant 2\left| \sum_{j=1}^{N} (\alpha_{G_k j} - \alpha_{ij}^\star) \right| + 2\lambda \sqrt{\sum_{j=1}^{N} \frac{(\alpha_{G_k j} - \alpha_{ij}^\star)^2}{m_j}} \\
&\leqslant 2\|\boldsymbol{\alpha}_{G_k} - \boldsymbol{\alpha}_i^\star\|_1 + 2\lambda\|\boldsymbol{\alpha}_{G_k} - \boldsymbol{\alpha}_i^\star\|_2 \\
&\leqslant O\left( \eta(1-\tau)\sqrt{\frac{N}{N-1}} \right).
\end{aligned}$$

$\square$

# D    Proof of Theorem 4 and Lemma 3

First we prove Lemma 3.

*Proof.* When $\{\boldsymbol{\alpha}_1^\star, \ldots, \boldsymbol{\alpha}_N^\star\}$ satisfy the $(1 + \gamma, \epsilon)$-approximation-stability property, let $\mathcal{P}^\star = \{P_1^\star, \ldots, P_K^\star\}$ be the optimal group partition with the minimum $\Phi(\mathcal{P})$. The average distance in $\mathcal{P}^\star$ is defined as $\bar{d} = \frac{1}{N}\mathrm{OPT}_{\Phi(\mathcal{P})}$. For any constant $t > 2$, lemma 3 in Balcan et al. (2009) reveals that there are less than $6\epsilon N$ clients with $d_2(\boldsymbol{\alpha}_i^\star) - d_1(\boldsymbol{\alpha}_i^\star) \leqslant \frac{\gamma \bar{d}}{2\epsilon}$ and less than $\frac{t\epsilon N}{\gamma}$ clientss with $d_1(\boldsymbol{\alpha}_i^\star) \geqslant \frac{\gamma \bar{d}}{t\epsilon}$ in $\mathcal{P}^\star$. Recall that the critical distance in $\mathcal{P}^\star$ is defined as $d^\star = \frac{\gamma \bar{d}}{t\epsilon}$. In Algorithm 2, we run a constant-factor $K$-median approximation algorithm on $\{\boldsymbol{\alpha}_1^\star, \ldots, \boldsymbol{\alpha}_N^\star\}$ to compute an estimate $\hat{d} \in [\bar{d}, \beta \bar{d}]$ where $\beta > 1$ is a given constant. Let $\mathcal{P} = \{P_1, \ldots, P_K\}$ be the group partition returned by the DIVIDE part in Algorithm 2. Note that the critical distance in $\mathcal{P}$ is defined as $\hat{d}^\star = \frac{\gamma \hat{d}}{\beta t\epsilon} \leqslant \frac{\gamma \bar{d}}{t\epsilon} = d^\star$. We have that the set $\mathcal{B} = \{\boldsymbol{\alpha}_i^\star \in S \mid d_1(\boldsymbol{\alpha}_i^\star) \geqslant \hat{d}^\star \vee d_2(\boldsymbol{\alpha}_i^\star) - d_1(\boldsymbol{\alpha}_i^\star) \leqslant \frac{t}{2}\hat{d}^\star\}$ of bad clients in $\hat{\mathcal{P}}$ has size $|\mathcal{B}| < (6 + \frac{t}{\gamma})\beta\epsilon N$. $\qquad\square$

Then we prove Theorem 4.

*Proof.* For good client $C_i$ in group $P_k$, $d_1(\boldsymbol{\alpha}_i^\star) \leqslant \hat{d}^\star = \frac{\gamma \hat{d}}{\beta t\epsilon} \leqslant \frac{\gamma \bar{d}}{t\epsilon}$. According to the equivalence of norms in the normed vector spaces, we assume $d(\cdot, \cdot) = \|\cdot\|_1$. Notice the following inequality:

$$\|\boldsymbol{x}\|_2 = \left(\sum_{i=1}^N |x_i|^2\right)^{\frac{1}{2}} \leq \sum_{i=1}^N \left(|x_i|^2\right)^{\frac{1}{2}} = \sum_{i=1}^N |x_i| = \|\boldsymbol{x}\|_1$$

Let $\boldsymbol{\alpha}_{P_k}$ be the average collaboration vector that Algorithm 2 uses to train model $h_{\boldsymbol{\alpha}_{P_k}}$ for group $P_k$. Then we get the required result

$$upp(\hat{h}_{P_k}) - upp(\hat{h}_{\boldsymbol{\alpha}_i^\star}) = 2\sum_{j=1}^N \left(\alpha_{P_k j} - \alpha_{ij}^\star\right) d_{\mathcal{H}}(\mathcal{D}_i, \mathcal{D}_j) + 2\lambda \left(\sqrt{\sum_{j=1}^N \frac{\alpha_{P_k j}^2}{m_j}} - \sqrt{\sum_{j=1}^N \frac{\alpha_{ij}^{\star 2}}{m_j}}\right)$$

$$\leqslant 2\left|\sum_{j=1}^N (\alpha_{P_k j} - \alpha_{ij}^\star)\right| + 2\lambda \sqrt{\sum_{j=1}^N \frac{(\alpha_{P_k j} - \alpha_{ij}^\star)^2}{m_j}}$$

$$\leqslant 2\|\boldsymbol{\alpha}_{P_k} - \boldsymbol{\alpha}_i^\star\|_1 + 2\lambda\|\boldsymbol{\alpha}_{P_k} - \boldsymbol{\alpha}_i^\star\|_2$$

$$\leqslant (2 + 2\lambda)\|\boldsymbol{\alpha}_{P_k} - \boldsymbol{\alpha}_i^\star\|_1$$

$$\leqslant O\left(\frac{\gamma \mathrm{OPT}_{\Phi(\mathcal{P})}}{\epsilon t N}\right).$$

$\qquad\square$

# E  Proof of the relationships between two divergences

We begin by proving some useful lemmas.

Let $(\mathcal{X}, \mathcal{A})$ be a measurable space. Let $P$ and $Q$ be two probability measures on $(\mathcal{X}, \mathcal{A})$. Suppose that $\nu$ is a $\sigma$-finite measure on $(\mathcal{X}, \mathcal{A})$ satisfying $P \ll \nu$ and $Q \ll \nu$. Define $p = dP/d\nu$, and $q = dQ/d\nu$. The total variation distance between $P$ and $Q$ is defined as follows:

$$V(P, Q) = \sup_{A \in \mathcal{A}} |P(A) - Q(A)| = \sup_{A \in \mathcal{A}} \left|\int_A (p - q)d\nu\right|.$$

It is easy to prove that $V(P, Q)$ satisfies the axioms of distance and $0 \leq V(P, Q) \leq 1$. In this section, we will often write for brevity $\int(\ldots)$ instead of $\int(\ldots)d\nu$ for simplicity.

**Lemma 6.**

$$V(P, Q) = \frac{1}{2}\int |p - q|d\nu = 1 - \int \min(p, q)d\nu.$$

*Proof.* Denote $A_0 = \{x \in \mathcal{X} : q(x) \geq p(x)\}$. Then we get $\int |p - q| d\nu = 2\int_{A_0}(q - p)d\nu$ and

$$V(P,Q) \geq Q(A_0) - P(A_0) = \frac{1}{2}\int |p - q|d\nu = 1 - \int \min(p, q)d\nu.$$

On the other hand, for all $A \in \mathcal{A}$ we have

$$\left|\int_A (q - p)d\nu\right| = \left|\int_{A \cap A_0}(q - p)d\nu + \int_{A \cap A_0^c}(q - p)d\nu\right|$$

$$\leq \max\left\{\int_{A_0}(q - p)d\nu, \int_{A_0^c}(p - q)d\nu\right\} = \frac{1}{2}\int |p - q|d\nu$$

where $A_0^c$ is the complement of $A_0$. Then $V(P,Q) = Q(A_0) - P(A_0)$ implies the required result. $\square$

**Lemma 7.**

$$\int \min(p, q)d\nu \geq \frac{1}{2}\left(\int \sqrt{dPdQ}\right)^2.$$

*Proof.* By noticing that $\int \max(p, q) + \int \min(p, q) = 2$, we obtain

$$\left(\int \sqrt{pq}\right)^2 = \left(\int \sqrt{\min(p, q)\max(p, q)}\right)^2 \leq \int \min(p, q)\int \max(p, q)$$

$$= \int \min(p, q)\left[2 - \int \min(p, q)\right] \leq 2\int \min(p, q)$$

which proves the required inequality. $\square$

**Lemma 8.**

$$\int \min(p, q)d\nu \geq \frac{1}{2}\exp(-d_{\mathrm{KL}}(P\|Q)).$$

*where $d_{\mathrm{KL}}(P\|Q)$ is the Kullback–Leibler (KL) divergence.*

*Proof.* It is sufficient to assume that $d_{\mathrm{KL}}(P\|Q) < +\infty$. Using the Jensen inequality we get

$$\left(\int \sqrt{pq}\right)^2 = \exp\left(2\log\int_{pq>0}\sqrt{pq}\right) = \exp\left(2\log\int_{pq>0}p\sqrt{\frac{q}{p}}\right)$$

$$\geq \exp\left(2\int_{pq>0}p\log\sqrt{\frac{q}{p}}\right) = \exp(d_{\mathrm{KL}}(P\|Q)).$$

By comparing this result with that in Lemma 7, we yield the required result. $\square$

Now we prove the result in our paper.

*Proof.* When the hypothesis space $\mathcal{H}$ is the class of functions taking values in $[-1, 1]$, the Integral Probability Metrics (IPM) $d_{\mathcal{H}}(\mathcal{D}_i, \mathcal{D}_j) = \sup_{h \in \mathcal{H}}|\mathcal{L}_{\mathcal{D}_i}(h) - \mathcal{L}_{\mathcal{D}_j}(h)|$ can also be viewed as the total variation distance. According to Pinsker's inequality we have

$$d_{\mathcal{H}}(\mathcal{D}_i, \mathcal{D}_j) \leqslant \sqrt{\frac{d_{\mathrm{KL}}(\mathcal{D}_i\|\mathcal{D}_j)}{2}},$$

where $d_{\mathrm{KL}}(\mathcal{D}_i, \mathcal{D}_j)$ is the Kullback–Leibler (KL) divergence. We can get the following result by noticing that $d_{\mathrm{JS}}(\mathcal{D}_i\|\mathcal{D}_j) = \frac{1}{2}d_{\mathrm{KL}}(\mathcal{D}_i\|\mathcal{D}_j) + \frac{1}{2}d_{\mathrm{KL}}(\mathcal{D}_j\|\mathcal{D}_i)$ where $d_{\mathrm{JS}}(\mathcal{D}_i\|\mathcal{D}_j)$ is the Jensen–Shannon (JS) divergence

$$2d_{\mathcal{H}}(\mathcal{D}_i, \mathcal{D}_j)^2 \leqslant \frac{d_{\mathrm{KL}}(\mathcal{D}_i\|\mathcal{D}_j)}{2} + \frac{d_{\mathrm{KL}}(\mathcal{D}_j\|\mathcal{D}_i)}{2} \iff d_{\mathcal{H}}(\mathcal{D}_i, \mathcal{D}_j) \leqslant \sqrt{\frac{d_{\mathrm{JS}}(\mathcal{D}_i\|\mathcal{D}_j)}{2}}.$$

By combining Lemma 6 and Lemma 8, we can easily obtain that

$$d_{\mathcal{H}}(\mathcal{D}_i, \mathcal{D}_j) \leq 1 - \frac{1}{2}\exp(-d_{\mathrm{KL}}(\mathcal{D}_i\|\mathcal{D}_j)).$$

Notice that

$$-\log\left(2 - 2d_{\mathcal{H}}(\mathcal{D}_i, \mathcal{D}_j)\right) \leqslant \frac{d_{\mathrm{KL}}(\mathcal{D}_i\|\mathcal{D}_j)}{2} + \frac{d_{\mathrm{KL}}(\mathcal{D}_j\|\mathcal{D}_i)}{2}$$

$$\iff d_{\mathcal{H}}(\mathcal{D}_i, \mathcal{D}_j) \leq 1 - \frac{1}{2}\exp(-d_{\mathrm{JS}}(\mathcal{D}_i\|\mathcal{D}_j)).$$

Thus we get the required result

$$d_{\mathcal{H}}(\mathcal{D}_i, \mathcal{D}_j) \leqslant min\left\{1 - \frac{1}{2}e^{-d_{\mathrm{JS}}(\mathcal{D}_i\|\mathcal{D}_j)}, \sqrt{\frac{d_{\mathrm{JS}}(\mathcal{D}_i\|\mathcal{D}_j)}{2}}\right\}.$$

$\square$