# OpenReview forum: "Collaborative Learning by Detecting Collaboration Partners"
_NeurIPS.cc/2022/Conference — NeurIPS 2022 Accept_

### Official Review · Reviewer_XfiP · 2022-07-08

**Rating:** 5
**Confidence:** 4
**Soundness:** 3 good
**Presentation:** 3 good
**Contribution:** 2 fair

**Summary:**

This paper studies the collaborative learning problem, in which different clients could have non-i.i.d data. Compared with existing global or personalized methods, this paper proposes to detect collaboration partners to achieve a better collaboration. The authors formalize this problem as measuring the similarity between clients and finding an optimal partition of the collaboration federated network. The authors give various theoretical analysis and provide experimental results to verify the effectiveness of the proposed method.

**Questions:**

Please see weaknesses.

**Ethics Review Area:**

["I don’t know"]

**Limitations:**

The authors claim that they describe the limitations, but I did not found.

**Strengths And Weaknesses:**

## Strengths:
1.This paper considers a challenging problem, the collaboration among multiple clients, which is very useful and significant in reality.

2.This paper mainly focuses on theoretical analysis, including the similarity measurement between clients, the optimization of the partition, and the error bound of the approximated partition;

## Weakness:
### Motivation:
1. The motivation in this paper is to find a partition for the collaboration network. It is reasonable and efficient. However, some related work and more discussions are missing, for example [1,2].

### Theory:
1.Some theoretical results I’m very familiar with, e.g., theorem 1 and theorem 2. I’m afraid they are quite similar to existing theoretical work in domain adaptation and domain generalization. For the other theoretical conclusions, the authors also seems to refer to existing work. That being said, it’s not a big problem. More discussions about the difference between the theoretical analysis and existing work are necessary;

### Experiments:
The experiments in this paper are seriously insufficient.
1.the idea in this paper is clustering similar or helpful clients. More baselines are necessary, for example [2].
2.the experiments focus on two image datasets. More other data are necessary.

### Privacy:
This paper considers the problem of collaborative learning. The proposed method needs to measure the similarity between clients, which could cause privacy leakage in a federated learning setting. Could the authors discuss more about the privacy concerns?

[1] Cui S, Liang J, Pan W, et al. Learning to Collaborate[J]. arXiv preprint arXiv:2108.07926, 2021.

[2]. Ghosh A, Chung J, Yin D, et al. An efficient framework for clustered federated learning[J]. Advances in Neural Information Processing Systems, 2020, 33: 19586-19597.

---

> ### Author Response · Authors · 2022-08-02
> **Authors' Response (Part 1)**
>
> Thank you for your efforts and suggestions. Please refer to our reply below.
>
> **Q1: The motivation in this paper is to find a partition for the collaboration network. It is reasonable and efficient. However, some related work and more discussions are missing.**
>
> A1: Thank you for your suggestions. We will add more discussions about the related work in the revised version.
>
> **Q2: Some theoretical results I’m very familiar with, e.g., theorem 1 and theorem 2. I’m afraid they are quite similar to existing theoretical work in domain adaptation and domain generalization. For the other theoretical conclusions, the authors also seems to refer to existing work. That being said, it’s not a big problem. More discussions about the difference between the theoretical analysis and existing work are necessary;**
>
> A2: As discussed by Avrim Blum et al. in their theoretical study on collaborative learning [1], collaborative learning is incomparable to domain adaptation since the goal of collaborative learning is to learn good classifiers on each individual client. Previous theoretical studies on collaborative learning focus on learning the centralized model or personalized models [1, 2, 3, 4], while we focus on learning $K$ models which have comparable performance to the personalized models by detecting collaboration partners. Since the collaboration relationship is complicated, we construct the incoming-edge similarity matrix $\mathbf{U}$ and use modularity maximization to detect collaboration partners. Most importantly, we prove that $K$ models can be learned by detecting collaboration partners and these $K$ models are good approximations of the personalized models. We will make this clear in the revised version.
>
> **References**
>
> [1] Avrim Blum, Nika Haghtalab, Ariel D. Procaccia, et al. Collaborative PAC learning. In NIPS 2017.
>
> [2] Jiecao Chen, Qin Zhang, and Yuan Zhou. Tight bounds for collaborative PAC learning via multiplicative weights. In NeurIPS 2018.
>
> [3] Huy L. Nguyen and Lydia Zakynthinou. Improved algorithms for collaborative PAC learning. In NeurIPS 2018.
>
> [4] Mehryar Mohri, Gary Sivek, and Ananda Theertha Suresh. Agnostic federated learning. In ICML 2019.
>
> **Due to space limitations, we leave the rest of our response to the next comment.**

---

> > ### Author Response · Authors · 2022-08-02
> > **Authors' Response (Part 2)**
> >
> > **Q3: The experiments in this paper are seriously insufficient. 1.the idea in this paper is clustering similar or helpful** **clients. More baselines are necessary, for example. 2.the experiments focus on two image datasets. More other** **data are necessary.**
> >
> > A3: Our paper mainly focuses on theoretical study on collaborative learning and we prove that $K$ models can be learned by detecting collaboration partners and these $K$ models are good approximations of the personalized models. With the collaboration partners detected in our paper, various collaborative learning methods can be adopted to learn the models. We believe that our paper is beneficial to the collaborative learning community and can provide theoretical guarantee for designing algorithms. Designing new algorithms for learning the model is beyond the scope of our paper, and the experiments in our paper are conducted to verify our theoretical findings. In the paper of [5] you mentioned, they proposed the method IFCA that alternately estimates the cluster identities of clients and optimizes model parameters for the client clusters. Although they provided theoretical analysis, it is based on the strong assumption that the models are linear. From the theoretical perspective, our result has the advantage over the result in [5]. Moreover, the result in [5] also relies on good initializations.
> >
> > We use the source code released by [5] on Github and run the method IFCA in [5] on MNIST and CIFAR-10 following the same client partitioning and network structures as that in our paper. For the general case, *i.e.*, the clients don't have clustering structure, the accuracy (%) of IFCA in [5] on 10 clients on MNIST is (95.6, 96.7, 85.5, 88.6, 89.8, 89.4, 88.9, 93.8, 88.2, 87.1). For the case that the clients have clustering structure, the accuracy (%) of IFCA on 10 clients on MNIST is (95.8, 96.3, 91.2, 94.3, 93.1, 89.9, 87.7, 96.2, 88.8, 92.2). From the results, we find that IFCA in [5] performs worse than ours on MNIST. Due to limited time, we only run the experiments on CIFAR-10 for the general case, *i.e.*, the clients don't have clustering structure. The accuracy (%) of IFCA in [5] on 20 clients on CIFAR-10 is (48.4, 50.6, 47.8, 57.3, 51.2, 45.9, 42.1, 57.3, 47.1, 45.1, 43.5, 59.2, 46.1, 47.5, 54.3, 46.9, 49.2, 38.7, 64.1, 52.8), which is much worse than that of ours on CIFAR-10. Since IFCA in [5] is based on clustering, for the general case, *i.e.*, the clients don't have clustering structure, the performance of IFCA is poor. As mentioned in [5], IFCA relies on good initializations.  We didn't repeat the experiments for multiple times due to time limitations.
> >
> > We will verify our theoretical findings on more datasets in the revised version, *e.g.*, synthetic data and FEMNIST.
> >
> > **Q4: This paper considers the problem of collaborative learning. The proposed method needs to measure the** **similarity between clients, which could cause privacy leakage in a federated learning setting. Could the authors** **discuss more about the privacy concerns?**
> >
> > A4: In the federated learning setting, it is also necessary to consider the similarity between clients, but it does not mean that the clients have to transfer their data. For privacy concerns, the similarity can be estimated through distribution statistics [6], activation vectors [7], etc.
> >
> > **References**
> >
> > [5] Ghosh A, Chung J, Yin D, et al. An efficient framework for clustered federated learning. In NeurIPS 2020.
> >
> > [6] Mi Luo, Fei Chen, Dapeng Hu, et al. No fear of heterogeneity: Classifier calibration for federated learning with non-iid data. In NeurIPS 2021.
> >
> > [7] Wei Chen, Kartikeya Bhardwaj, and Radu Marculescu. Fedmax: Mitigating activation divergence for accurate and communication-efficient federated learning. In ECML-PKDD 2020.

---

### Official Review · Reviewer_VDbb · 2022-07-11

**Rating:** 7
**Confidence:** 2
**Soundness:** 3 good
**Presentation:** 3 good
**Contribution:** 3 good

**Summary:**

This paper studied collaborative learning for heterogeneous clients. To achieve personalization while avoiding training personalized models for a large number of clients, the authors proposed methods to detect collaboration partners and adaptively learn models for client groups. The authors provide a theoretical guarantee that the expected risk of the learned group model is close to the personalized model. Empirical results on real-world datasets validated the theoretical result showing group model is a good approximation of the personalized model.

**Questions:**

Could the authors compare Theorem 3 and Theorem 4 under the same assumption, and discuss how much tighter Theorem 4 could be?
Will ACLC perform better than ACLMM following the setting of Figure 5, especially with large $N$?

**Limitations:**

See weakness.

**Strengths And Weaknesses:**

Strength:
The motivation of learning models for client clusters to balance between computation and personalization is intuitive. The two algorithms, ACLMM which applies Modularity Maximization to estimate client group, and ACLC which assumes intrinsic client clustering, both are analyzed with generalization bounds.

Weakness:
I only have mostly minor concerns. While the two proposed algorithms are theoretically analyzed, they are not directly compared. It seems that with the $(1 + \gamma, \epsilon)$-approximation-stability assumption in Theorem 4, ACLC has a better generalization bound than ACLMM (when $OPT_$\Phi(P)$ is small) and it would be interesting to compare the bounds in detail. It is also worth empirically testing ACLMM in Figure 5.

---

> ### Author Response · Authors · 2022-08-02
> **Authors' Response**
>
> Thank you for your efforts and suggestions. Please refer to our reply below.
>
> **Q1: Could the authors compare Theorem 3 and Theorem 4 under the same assumption, and discuss how much tighter Theorem 4 could be? Will ACLC perform better than ACLMM following the setting of Figure 5, especially with large N?**
>
> A1: For the general case, *i.e.*, the clients don't have clustering structure and the $(1+\gamma, \epsilon)$-approximation-stability property doesn't hold, the result in Theorem 4 cannot be derived. For the case that the clients have clustering structure: in Theorem 3, $\tau$ is the minimum weight of intra-group strong edges where $\tau \leqslant 1$ and is in the order of $\frac{1}{N}$, while $\eta$ is the maximum length of the shortest strong-paths in all groups and increases with $N$; in Theorem 4, $\gamma$ and $\epsilon$ are constants and $\mathrm{OPT}_{\Phi(\mathcal{P})}$ is close to zero due to the clustering structure. Thus, for the case that the clients have clustering structure, the result in Theorem 4 is tighter than that in Theorem 3. Following the setting of Figure 5 with large $N$, ACLC performs better than ACLMM since the result in Theorem 3 implies that the performance of ACLMM is worse than that of ACLC with large $N$.

---

### Official Review · Reviewer_br4T · 2022-07-13

**Rating:** 6
**Confidence:** 2
**Soundness:** 3 good
**Presentation:** 2 fair
**Contribution:** 3 good

**Summary:**

This paper studies the collaborative learning in personalized setting, which considers the heterogeneous distribution among clients. The paper proposes a method based on graph partition to learn personalized models for each partition group, which is shared by all clients within one partition group. The paper first provides the generalization bound for each client, from which the optimal personalized model is derived via optimizing on the contribution from the dataset of each client. Then, a contribution graph is constructed from the contribution weight, and similar clients are partitioned into one group. After filtering out bad clients, the personalized model for one group is learned based on the average weights of the clients within the group. The paper provides theoretically analysis on the increase of the upper bound of the excess risk due to the relaxation from optimal personalized model to group personalized model. Experiments demonstrate the group personalized model has similar performance with the optimal personalized model and justifies the proposed method.


**Questions:**

1.One crucial step is to calculate the IPM between different data distributions. It is estimated by Eq 8, which involves the JS divergence between two data distributions. So how do you calculate the JS divergence between two datasets of clients.

2.It seems that model training of one group needs the datasets from all clients. Does that mean clients need to transfer their local datasets? It seems that if sharing gradients, each client is required to keep all models of all groups and calculate gradients K(number of groups) times.




**Limitations:**

The paper does not discuss limitations of the work. It provides the discussion on social impact.


**Strengths And Weaknesses:**

This paper has several merits. First, the theoretical part is clear, rigorous, and sound. It justifies the effectiveness of the proposed framework. Second, the proposed method, which is based on graph partition of client contribution, is novel and interesting to me.

However, I am not very familiar with this research topic. My major concern is on the writing of this paper, as it is sometimes hard to follow. For example, there are no subsections within Section 4 "Our Work", but to my understanding this section first talks about the optimal model, then provides graph partition based learning algorithm, and last gives tighter bounds with further assumptions. Thus, I think section4 can be separated into at least 3 subsections. The authors are encouraged to improve the presentation of this work. I have some more questions to ask the authors. Please see the comment below.

---

> ### Author Response · Authors · 2022-08-02
> **Authors' Response**
>
> Thank you for your efforts and suggestions. Please refer to our reply below.
>
> **Q1: My major concern is on the writing of this paper, as it is sometimes hard to follow ... I think section 4 can be separated into at least 3 subsections.**
>
> A1: Thank you for your suggestions. We will reorganize this paper and separate Section 4 into 3 subsections in the revised version.
>
> **Q2: One crucial step is to calculate the IPM between different data distributions. It is estimated by Eq.8, which involves the JS divergence between two data distributions. So how do you calculate the JS divergence between two datasets of clients.**
>
> A2: The JS divergence is defined as $d_{\mathrm{JS}}(P||Q) = \frac{1}{2} \sum p(x) \log \left(\frac{p(x)}{\frac{p(x)+q(x)}{2}}\right)+\frac{1}{2} \sum q(x) \log \left(\frac{q(x)}{\frac{p(x)+q(x)}{2}}\right) $, where $P, Q$ are distributions. For the discriminative models with two samples $S_p \in D_{p}, S_q \in D_{q}$ , the JS divergence can be calculated as $d_{\mathrm{JS}}(S_p||S_q) = d_{\mathrm{JS}}(P(y^p|\mathbf{x}^p,h^p), P(y^q|\mathbf{x}^q,h^q))$ [1] where $P(y^p|\mathbf{x}^p,h^p)$ is the conditional probability generated with discriminative model $h^p$. Thus we can use the conditional distribution generated with discriminative model $h$ to estimate the JS divergence.
>
> **Q3: It seems that model training of one group needs the datasets from all clients. Does that mean clients need to transfer their local datasets? It seems that if sharing gradients, each client is required to keep all models of all groups and calculate gradients K(number of groups) times.**
>
> A3: Training the model of one group needs the data information from all clients, but it does not mean the clients have to transfer their local datasets. Label information, distribution statistics and gradient information can be used to capture the data information [2, 3]. If sharing gradients, for learning personalized models each client needs to calculate gradients for $N$ times, while for learning $K$ models each client only needs to calculate gradients for $K$ times. These $K$ models have comparable performance to the personalized models and $K \ll N$.
>
> **References**
>
> [1] Xiao Li, Jeff Bilmes. A Bayesian Divergence Prior for Classiffier Adaptation. In AISTATS 2007.
>
> [2] Mi Luo, Fei Chen, Dapeng Hu, et al. No fear of heterogeneity: Classifier calibration for federated learning with non-iid data. In NeurIPS 2021.
>
> [3] Xinyi Xu, Lingjuan Lyu, Xingjun Ma, et al. Gradient driven rewards to guarantee fairness in collaborative machine learning. In NeurIPS 2021.

---

### Official Review · Reviewer_DG5y · 2022-07-26

**Rating:** 5
**Confidence:** 2
**Soundness:** 3 good
**Presentation:** 2 fair
**Contribution:** 3 good

**Summary:**

This paper introduces a novel approach to partition the clients into $K$ groups and build personalized model for each group. The approach leverage graph clustering method for learning purpose. The paper also provided a list of learning bounds. The experimental results demonstrate the proposed $K$ models can produce learning results close to personalized models.



**Questions:**

1. Can some greedy approach be applied here? like greedy clustering (merging the most similar pairs, or something along this line)

2. Is Lemma 1 essential? What about using other simple/alternative approaches for optimizing modularity (as this is a well studied problem)

**Limitations:**

Yes

**Strengths And Weaknesses:**

Strengths:

1. The problem is interestingly and timely. It can potentially be applied to many applications
2. The approach is based on graph theoretical results
3. The paper provides rather rigorous theoretical analysis

Weaknesses:
1. The writing can be improved and better organized
2. Lack of base lines for comparison; the experimental results are rather weak.
3. Some Figures are too small to read (like those in Figure 5)

---

> ### Author Response · Authors · 2022-08-02
> **Authors' Response**
>
> Thank you for your efforts and suggestions. Please refer to our reply below.
>
> **Q1: The writing can be improved and better organized**
>
> A1: Thank you for your suggestions. We will reorganize this paper and improve the writing in the revised version.
>
> **Q2: Lack of baselines for comparison; the experimental results are rather weak**
>
> A2: Collaborative learning is a paradigm that allows many clients to learn models through collaboration. It is first theoretically formalized by Avrim Blum et al. [1], where they considered the centralized setting and the personalized setting and proved the superiority of collaborative learning. Some previous works followed this direction and studied collaborative learning theoretically [2, 3, 4]. [2, 3] improved the results in [1], while [4] provided upper bounds for the centralized model. In real-world applications where there are numerous heterogeneous clients, the centralized model may perform badly on some clients while learning a personalized model for each client costs unaffordable computational resources. A reasonable way is to learn $K$ models that have comparable performance to the personalized models.
>
> Unfortunately, few previous works focus on this challenging theoretical problem. The main contribution of this paper is: we prove that $K$ models can be learned by detecting collaboration partners and these $K$ models are good approximations of the personalized models. With the collaboration partners detected in our paper, various collaborative learning methods can be adopted to learn the models. We believe that our paper is beneficial to the collaborative learning community and can provide theoretical guarantee for designing algorithms. Designing new algorithms for learning the model is beyond the scope of our paper, and the experiments are conducted to verify our theoretical findings.
>
> **Q3: Some Figures are too small to read (like those in Figure 5)**
>
> A3: Thank you for your suggestions. We will adjust the figure size in the revised version.
>
> **Q4: Can some greedy approach be applied here? like greedy clustering (merging the most similar pairs, or** **something along this line).**
>
> A4: Our paper is a theoretical paper, where we prove $K$ models which have comparable performance to the personalized models can be learned by detecting collaboration partners. The greedy methods can also be applied, but the collaboration partners detected by these greedy methods may not have theoretical guarantee.
>
> **Q5: Is Lemma 1 essential? What about using other simple/alternative approaches for optimizing modularity (as** **this is a well studied problem).**
>
> A5: As mentioned above, we mainly focus on the theoretical study on collaborative learning. We use SDP to solve modularity maximization with the theoretical guarantee provided by lemma 1. Other methods for optimizing modularity can also be considered here, but most of these methods are heuristic without theoretical guarantee. That’s why we use SDP based on lemma 1 in our work.
>
> **References**
>
> [1]Avrim Blum, Nika Haghtalab, Ariel D. Procaccia, et al. Collaborative PAC learning. In NIPS 2017.
>
> [2] Jiecao Chen, Qin Zhang, and Yuan Zhou. Tight bounds for collaborative PAC learning via multiplicative weights. In NeurIPS 2018.
>
> [3] Huy L. Nguyen and Lydia Zakynthinou. Improved algorithms for collaborative PAC learning. In NeurIPS 2018.
>
> [4] Mehryar Mohri, Gary Sivek, and Ananda Theertha Suresh. Agnostic federated learning. In ICML 2019.

---

### Meta-Review · Area_Chair_ybne · 2022-08-26

**Recommendation:** Accept
**Confidence:** Certain

**Metareview:**

In this submission, the authors propose a novel method to handle the heterogeneous clients challenge in FL by detecting collaboration partners and adaptively learning models for client groups. The authors provide the theoretical analysis of the error bound of the approximated partition, which justifies the effectiveness of the proposed method. The experimental results on the real-world dataset also validate that the group model is a good approximation of the personalized model. Due to these, I recommend accepting this submission.

However, I do have one major concern about the insufficiency of the experiment (as pointed out by reviewer XfiP). More datasets and baselines should be adopted.

Further, this submission also can be improved based on the comments from all the reviewers and the discussion between reviewers and authors. Hope they find these useful, and make this submission a better one.


**Award:**

No

---

### Decision · Program_Chairs · 2022-09-14

Accept